# BuilderBench – A benchmark for generalist agents

## Abstract

Today's AI models learn primarily through mimicry and sharpening, so it is not surprising that they struggle to solve problems beyond the limits set by existing data. To solve novel problems, agents should acquire skills for exploring and learning through experience. Finding a scalable learning mechanism for developing agents that learn through interaction remains a major open problem. In this work, we introduce BuilderBench, a benchmark to accelerate research into agent pre-training that centers open-ended exploration. BuilderBench requires agents to learn how to build any structure using blocks. BuilderBench is equipped with (1) a hardware accelerated simulator of a robotic agent interacting with various physical blocks, and (2) a task-suite with over 42 diverse target structures that are carefully curated to test an understanding of physics, mathematics, and long-horizon planning. During training, agents have to explore and learn general principles about the environment without any external supervision. During evaluation, agents have to build the unseen target structures from the task suite. Solving these tasks requires a sort of *embodied reasoning* that is not reflected in words but rather in actions, experimenting with different strategies and piecing them together. Our experiments show that many of these tasks challenge the current iteration of algorithms. Hence, we also provide a "training wheels" protocol, in which agents are trained and evaluated to build a single target structure from the task suite. Finally, we provide single-file implementations of six different algorithms as a reference point for researchers.

*Can AI models build a world which today's generative models can only dream of?*

## 1 The need for a new benchmark

Today's artificial intelligence (AI) models acquire knowledge by combing through massive collections of human-generated data, enabling them to generate a wide array of images and write a wide array of stories. While this recipe has been highly successful in domains like vision and language, where models can learn from expert human photographers and writers, it is less clear how to apply this recipe to application areas that humans understand poorly today (e.g., biology, chemistry) (Ying et al., 2025; Silver & Sutton, 2025). Making progress will require that agents learn not only from human experience, but also from their own, self-collected experience. Agents will have to actively explore and run experiments to extract knowledge about the environment (Spelke & Kinzler, 2007). Agents will then have to consolidate this knowledge and use it to quickly solve novel tasks. Despite many works recognizing the importance of **open-ended exploration** and **learning through experience** (Stanley, 2017; Team, 2023), most benchmarks for building foundation models today focus on learning solely from human data.

This is not for lack of trying. There is a long line of interaction and exploration benchmarks built by researchers in reinforcement learning (RL) (Ecoffet et al., 2021; Tang et al., 2017), control (Plappert et al., 2018), and developmental robotics (Oudeyer et al., 2007). For example, maze navigation in ant-maze (Fu et al., 2021) or montezuma's revenge (Bellemare et al., 2013), or handful of tasks in the kitchen environment (Gupta et al., 2020). Other than a few exceptions like Minecraft (Guss et al., 2019), most widely used benchmarks only allow a handful of diverse behaviors (Rajeswar et al., 2023; Gupta et al., 2020; Fu et al., 2021; Tassa et al., 2018). Agents trained on even the most complex of these benchmarks (e.g., StarCraft (Vinyals et al., 2019), AI2Thor (Kolve et al., 2022), NetHack (Küttler et al., 2020)) do not seem to learn the same sort of common sense and

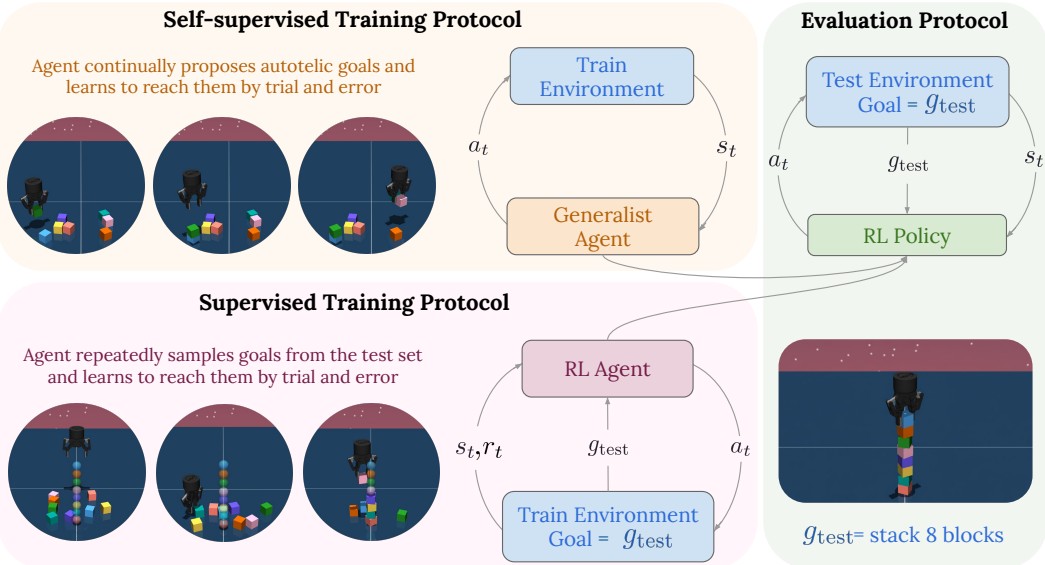

Figure 1: **The BuilderBench Benchmark.** *(Top Left)* Training consists of self-supervised exploration: agents can collect data to learn to reach various goals via trial and error. Agents are not given any information about the test-time goals or their distribution. *(Right)* During evaluation, the agent is given a goal and attempts to reach that goal by taking actions in the environment. *(Bottom Left)* For prototyping, we also include a "debug" mode where agents can learn to reach the test-time goals via trial and error.

reasoning skills that agents trained on human text do acquire (Wei et al., 2022). We argue that the key reason why, is that, there is not much that can be learned in the current generation of interactive benchmarks. Existing benchmarks rarely allow agents to practice skills ranging from exploration to prediction, from low-level control to high-level reasoning.

We envision a benchmark which enables an open-ended stream of interaction (Hughes et al., 2024; Sigaud et al., 2024), where training could only ever cover a tiny slice of all possible behaviors. In the same way that vision models today can paint pictures that go well beyond what is in their training data (e.g., an astronaut mowing the lawn), we envision embodied agentic systems that can solve tasks that go well beyond the tasks they have practiced solving before. Solving such a benchmark would require agents to have efficient exploration abilities. Moreover, it requires that exploration take into account an agent's generalization capabilities, since it will be impossible to perform all possible behaviors (Hughes et al., 2024). Agents should, in effect, become scientists, performing micro experiments in the environment to discover the laws governing the environment. Once these physical laws have been found, they can be used to make wide-ranging generalizations about how the entire environment works, and how one should act within it. *Our paper constructs an environment where such exploration is possible. One central insight of our paper is to show that this can actually be done using a surprisingly simple setup: block-building.*

**Why block-building?** Blocks conceptually form an atomic unit, allowing agents to build diverse structures. Many children spend years playing with blocks. Research in child development highlights that block play builds spatial (Reifel, 1984; Wexler et al., 1998; Casey et al., 2008; Singer et al., 2006) and arithmetic skills (Verdine et al., 2014; Cheng & Mix, 2014). In addition to being useful for early human cognitive development, block-building is a mathematically rich area[1] with a deep history in AI and planning (Gupta & Nau, 1992; Ahmad et al., 2019; Russell & Norvig, 2010). Building stable structures with blocks requires long-horizon planning and complex reasoning capabilities. While research on reasoning and generalization capabilities has almost become synonymous with large language models in recent years (Touvron et al., 2023; DeepSeek-AI, 2025; OpenAI, 2024; Team, 2025), block-building allows us to study whether this sort of reasoning and generalization can emerge from the ground up, through exploration and trial-and-error learning.

---

[1]In 2011, Paterson et al. (2007) was awarded the prestigious David P. Robbins Prize in mathematics for improving an upper bound on the maximum overhang using identical blocks.

To this end, we introduce the **BuilderBench** benchmark. BuilderBench is equipped with a fast simulator consisting of a robotic hand traversing in space and interacting with blocks, following Newtonian physics. This simple setup allows to design tasks which span several orders of magnitude of complexity. Tasks require motor skills like locomotion, grasping and throwing as well as higher-level skills such as logical reasoning (commutativity and associativity of pick and place ordering), geometrical reasoning (maximizing overhangs, packing problems) and intuitive physics (gravity, friction, toppling, balancing). Tasks also require reasoning about counterweights, buttresses, and temporary scaffolding. During training, agents must discover such skills through practice. During testing, agents must use those skills to build unseen structures (Figure 1). To succeed in building a large set of diverse structures, agents must learn general patterns of construction, rather than memorizing individual actions. Finally, one can easily scale the difficulty of tasks by increasing the number of blocks.

We summarize the main contributions of this paper below:

- We introduce BuilderBench, a benchmark that uses open-ended block-building to develop and evaluate agents for efficient exploration, reasoning and generalization abilities.

- The BuilderBench simulator is developed using MuJoCo (Todorov et al., 2012) and JAX (Bradbury et al., 2018). It is hardware accelerated and allows RL training between 10 to 100 times faster than purely CPU based open-ended benchmarks like Crafter (Hafner, 2022), Minecraft (Guss et al., 2019) or NetHack (Küttler et al., 2020) (see Appendix B for a speed test).

- We open-source BuilderBench, a task-suite of over 40 tasks to evaluate the performance of agents. Each task corresponds to a different block structure. Building each structure requires unique reasoning abilities.

- We open-source single-file implementations of four representative reinforcement learning (RL) algorithms and three self-supervised data-collection algorithms. Training runs are extremely fast (e.g., training a PPO agent to stack two blocks takes 30 minutes on a single GPU), reducing the barrier to entry for frontier RL research.

## 2 RELATED WORK.

AI benchmarks have driven progress in the field. Benchmarks such as MNIST (Deng, 2012), ImageNet (Russakovsky et al., 2015), Atari (Bellemare et al., 2013), Gym (Brockman et al., 2016), WMT (Chelba et al., 2014), SWE-bench (Jimenez et al., 2024), ARC-AGI (Chollet, 2019) have propelled research in deep learning, vision, RL and natural language processing. The aim of BuilderBench is to similarly propel research on RL agents. Below we discuss various aspects of this problem and prior attempts to tackle and benchmark them.

Reinforcement learning (RL) studies agents that learn through interaction. Standard RL benchmarks (Bellemare et al., 2013; Brockman et al., 2016; Tassa et al., 2018; Hafner, 2022; Küttler et al., 2020; Koyamada et al., 2023; Bonnet et al., 2024) have agents learn to maximize hand-designed rewards to solve a task of interest. These environments require agents to extract their own knowledge and novel solutions (e.g., endlessly bouncing the ball in breakout from DQN (Mnih et al., 2013) or the famous move 37 from AlphaGo (Silver et al., 2016)). However, these environments focus on solving a small range of tasks. As a result, RL agents typically possess narrow or poor generalization capabilities (Kirk et al., 2023). The type of generalization that is desired is not just towards perturbed observations or dynamics (Stone et al., 2021; Cobbe et al., 2020), but towards solving diverse unseen tasks (Ghosh et al., 2021).

Unsupervised RL is centered on devising objectives that let agents learn through trial and error without any rewards. Such methods usually try to learn generally useful skills (Gregor et al., 2016; Eysenbach et al., 2019) or collect exploratory data (Lee et al., 2020; Tang et al., 2017; Osband et al., 2016). But it is not clear how scalable these objectives are, mainly because the standard unsupervised RL benchmarks (Rajeswar et al., 2023; Fu et al., 2021; Tassa et al., 2018) contain only a handful of similar downstream tasks for evaluation. Hence, to properly evaluate generalization properties of agents, benchmarks need to have sufficiently complex and unseen test tasks.

Another set of methods that are closely related are ones which treat the problem of efficiently generalizing to unseen tasks as a learning problem itself. Meta-learning (Caruana, 1998; Finn et al., 2017; Schmidhuber, 1987) and few-shot learning (Vinyals et al., 2016; Snell et al., 2017) fall under this category. Initial progress was driven by benchmarks that arranged common supervised learning tasks episodically, testing how quickly models adapt to new tasks (Lake et al., 2015; Dhillon et al., 2020). Later work found that self-supervised pre-training on diverse datasets provided enough prior knowledge to directly solve most of the common supervised learning tasks (Radford et al., 2021; Brown et al., 2020; Devlin et al., 2018), blurring the boundary between memorizing prior knowledge and efficiently generalizing. We argue that open-ended domains and tasks are needed to disentangle the two. ARC-AGI (Chollet, 2019) uses the open-ended domain of discrete puzzles to measure a model's ability to efficiently use its priors. ARC tests models on a set of novel puzzles that require on-the-fly composition of a minimal set of core principles (Chollet, 2019; Spelke & Kinzler, 2007). BuilderBench is similarly structured. Solving tasks from the BuilderBench task-suite not only requires a concrete set of priors (e.g., an understanding of Newtonian physics), but requires using these priors to build unseen structures on-the-fly. Unlike ARC-AGI, where priors are directly provided through examples of solved puzzles, in BuilderBench agents have to discover priors on their own through interaction.

In addition to exploration and generalization, the BuilderBench task-suite highlights how block-building can also be used to evaluate various types of reasoning abilities (see Section 5.1 for details). Many of these abilities are typically studied only in isolation. For e.g., intuitive physics is evaluated in Chow et al. (2025); Riochet et al. (2020), motor skills in James et al. (2019); Melnik et al. (2021), planning in Valmeekam et al. (2023), mathematical reasoning in Lewkowycz et al. (2022); Ahn et al. (2024). In recent years, reasoning is almost exclusively studied using language models pretrained on data. However, BuilderBench allows us to evaluate and visualize reasoning that is not grounded in language and not learned using human data.

The most similar benchmarks to BuilderBench are recent benchmarks like Kinetix (Matthews et al., 2025), XLand (Team et al., 2021; Nikulin et al., 2023), and Minecraft (Guss et al., 2019). Kinetix provides a diverse set of rigid body tasks, constrained to 2D, to test zero shot generalization of agents. Tasks in Kinetix are procedurally generated. Unlike BuilderBench, these tasks do not clearly test diverse logical and mathematical reasoning abilities. XLand provides a vast set of multi-agent video-game like tasks, but is closed source and not readily available for academic research. Minecraft is a popular open-ended game that revolves around building various artifacts with blocks that has been used to develop generally capable agents from scratch (Hafner et al., 2024; Ma et al., 2022; Zhao et al., 2024). While based on the similar block-building foundations and an appealing benchmark, we believe BuilderBench is better suited for academic research due to the much faster speed of its simulator and an extensive carefully curated task-suite. Finally, BuilderBench is fully open source, making all of its components flexible and easy to adapt. In Appendix C, we have added more discussion on qualitative comparson of BuilderBench with prior benchmarks.

## 3 BUILDERBENCH - A BENCHMARK FOR GENERALIST AGENTS

This paper proposes the BuilderBench benchmark, which comes with task-suite of 42 tasks, where each task is a target block structure carefully curated for evaluating unique abilities. BuilderBench comes with a fast simulator consisting of an agent interacting with a varying number of blocks. In the following sections, we will describe the environment (Section 4), the task-suite (Section 5) and the training and evaluation protocols (Section 6).

## 4 BUILDERBENCH ENVIRONMENT.

The environment can be formulated as a Markov decision process (MDP) (Sutton & Barto, 2018), with states $s_t \in \mathcal{S}$ and actions $a_t \in \mathbf{A}$ and transition dynamics $T(s_{t+1} \mid s_t, a_t)$ and a maximum episode length $H$. An additional context parameter $n$ specifies the number of cube-shaped blocks in the environment. Each environment instance contains a single robot hand which can navigate in

3D space[2] and interact with the $n$ cubes. All interactions approximate real physics simulated using MuJoCo (Todorov et al., 2012).

**State space.** The observations include information about the arm and the cubes. For the arm, we include its global position coordinates ($\mathbb{R}^3$), orientation quaternion ($\mathbb{R}^4$), linear velocity ($\mathbb{R}^3$), and the distance between its two fingers ($\mathbb{R}$). For every cube, we include the global position coordinates ($\mathbb{R}^3$), orientation quaternion ($\mathbb{R}^4$), linear and angular velocity ($\mathbb{R}^6$).

**Action space.** The agent can manipulate its environment using a 5 dimensional action space. The first three actions control its position actuators, enabling navigation along the standard basis vectors. The fourth action controls the agent's yaw, enabling it to rotate about the global z-axis. The fifth action controls the finger actuators, which allows the agent to change its pinching width.

**Task specification.** Each task corresponds to a physically stable target structure built using cubes. To specify this structure, we provide a vector of target cube positions ($\mathbb{R}^{3k}$), where $k \leq n$ is the number of cubes in the target cube structure. This allows us to specify target structures that contain fewer cubes than the environment (see Figure 5 for an example).

As we will see in the next section, despite this seemingly simple setup, tasks can be arbitrarily complex and long-horizon. Qualitatively, we will see that solving tasks require multiple steps of high-level reasoning.

## 5 BUILDERBENCH TASK SUITE

The BuilderBench task suite contains a total of 42 tasks. In this section, we describe these tasks in detail and the design philosophy behind the task-suite. The task-suite is meant to be open-ended and address the challenges highlighted in Section 1. We start with a case-study of five different tasks from the BuilderBench task-suite, which is meant to showcase how each task requires the agent to unlock at least one distinct reasoning ability and compose various high-level skills sequentially. As described in Section 4, during evaluation, agents only have access to the positions of the masked cubes in the target structure. The complete list of tasks, along with visualizations and the capabilities required to solve them, is provided in Appendix E.

### 5.1 A CASE STUDY OF FIVE TASKS

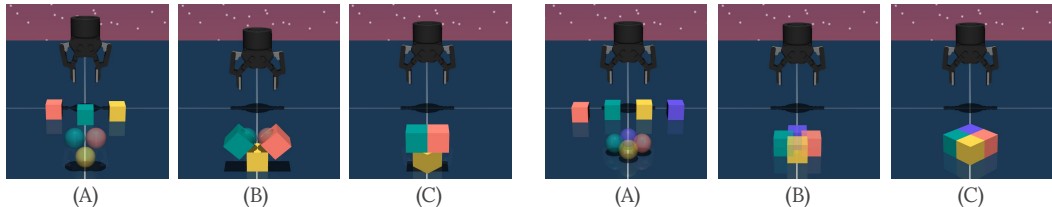

(A)  (B)  (C)          (A)  (B)  (C)

Figure 2: **T-Block** (Left) and **Four Cube Packing** (Right)

**Example 1: T-Block.** This task requires building a simple T shaped structure with one cube at the base, and two cubes on top (Figure 2). The second frame (B) shows what many people envision as the solution to this task. However, as show in the frame, this configuration isn't stable. Solving this task requires the reasoning insight to rotate the bottom cube by about $45°$. Since the diagonal of the cube's top surface is longer than its edge length, the rotated base provides sufficient support for both top cubes, enabling a stable T-shaped structure (see third frame).

**Example 2: Four Cube Packing.** This task tests geometric reasoning and spatial packing. The target structure is an arrangement of four cube centers placed at some distance along the four cardinal directions on the floor (see (A) of Figure 2). The distance is chosen such that the placement is impossible with the default cube orientation: the cubes overlap (see (B)). This results in a packing problem of arranging the cubes such that its centers form the target structure. To solve this, the agent needs to rotate each cube by $45°$ before placing it, which ensures the centers align correctly

---

[2]We do not include the entire robot because inverse kinematics is a solvable and orthogonal problem. This also significantly increases the reach of the robotic arm. This robot can be conceptually thought of as a crane.

without collision (see (C)). Due to the two fingered morphology of the robot, this task cannot be solved using pick and place primitives, but would require nudging the final block in place.

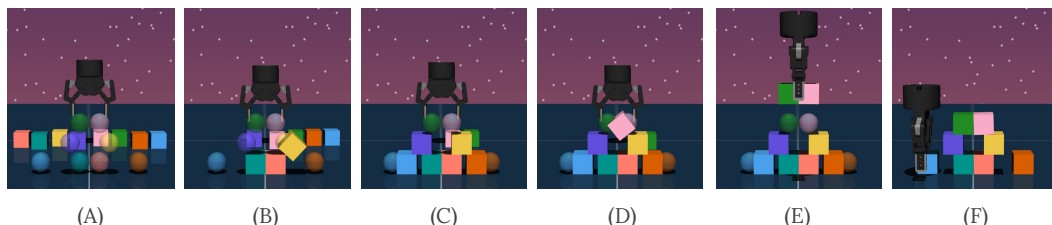

(A)      (B)      (C)      (D)      (E)      (F)

Figure 3: **Hexagonal Portal**

**Example 3: Hexagonal Portal.** This task requires constructing a hexagonal, portal-like structure using eight cubes and two extra cubes placed on the floor on either side. (Figure 3). A direct attempt to place the yellow or indigo cubes leads to toppling (see (B)). To stabilize them, the agent must first build two supporting scaffolds (see (C)). After the first two layers are in place, the pink and green cubes cannot be added independently since each would collapse inward (see (D)). Because no additional blocks remain, another scaffold cannot be used for support. Instead, the agent must discover a non-trivial maneuver – lifting and placing the pink and green cubes simultaneously (see (E)). Finally, the temporary orange and light-blue scaffolds must be carefully removed and placed in their desired position to complete the structure (see (F)). This task requires long-horizon planning and learning emergent skills like building scaffolds and learning to pick two cubes at once.

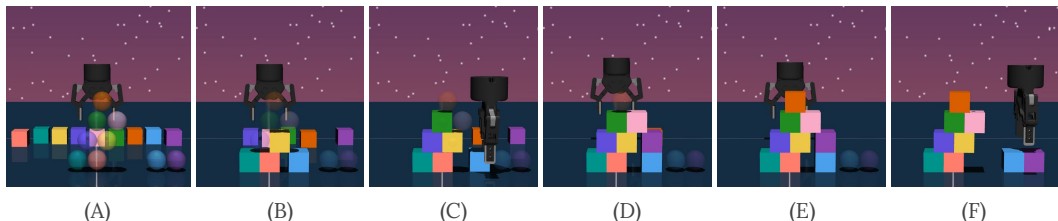

(A)      (B)      (C)      (D)      (E)      (F)

Figure 4: **Leaning Tower**

**Example 4: Leaning Tower.** The target is a leaning tower composed of seven blocks and two extra cubes placed on the floor (see (A) of Figure 4). Solving this task demands building two scaffolds and re-using the first one for the main tower. It also requires an understanding of the concept of counterweights for generating a stable overhang (an outward extension). The solution itself requires multiple steps of high level planning. After building the base, the yellow block in the second layer must be supported by a temporary scaffold (see blue cube in (B)). To stabilize the structure, the agent needs to add counterweights (the indigo and green cubes in (C)) and only then remove the scaffold (see (D)). To build the third and fourth layer, the agent has to build another set of scaffolds and counterweights. In particular, placing the pink block in the third layer requires a two-cube vertical scaffold (see (E)). Finally, the tower is completed by adding the counterweights (the green and orange cubes in (F)) and removing and repositioning the last scaffold (see (G)).

**Example 5: Maximum Overhang Problem.** In this task, the environment contains five cubes, but the task only specifies the target positions for three cubes (see (A) of Figure 5). But to put those three cubes in the target location, the agent will need to use all five blocks. To correctly place the green and the yellow cubes (whose target positions are not specified) in order to complete the task, the agent needs to solve the popular maximum overhang problem (see Paterson et al. (2007) for the solution). The main intuition is that at any level, the collective center of the mass of all the cubes above, should not be on the right of the level's boundary. Without such a placement, the task is impossible to solve. The pink cube is specified to "distract" the agent from simply holding the indigo cube in place.

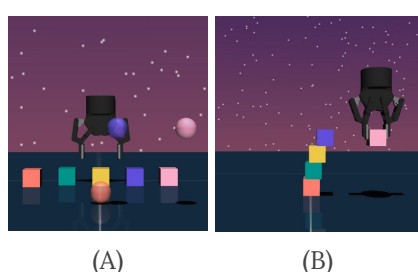

(A)      (B)

Figure 5: **Maximum overhang problem**

This case-study illustrates how a block-building setup with a handful of blocks can result in open-ended tasks that can be used to test high-level reasoning abilities. Agents which do not have access to these tasks have almost no chance of encountering them during training. For solving these tasks, we anticipate that agents would have to learn key knowledge priors via exploration (Spelke & Kinzler, 2007), and learn mechanisms to efficiently use them on the fly (Chollet, 2019). In the next section, we outline the general design principles that underlie the tasks in the BuilderBench task-suite.

### 5.2 DESIGN PHILOSOPHY BEHIND THE BUILDERBENCH TASK-SUITE.

The primary goal of the task-suite is to capture the main challenges in evaluating open-ended exploration and generalization (highlighted in Section 1) and provide a meaningful feedback signal for algorithmic research. To best support these goals, we followed the following design principles:

**Solving different tasks should require distinct skills.** For example, once an agent learns how to pick and place two blocks, extending this to three or more independent blocks does not qualitatively require an additional ability. We have designed tasks such that they demand a range of motor skills, including grasping, nudging, and throwing. Importantly, tasks also require logical reasoning skills, such as commutativity and associativity of blocks (pick and place ordering), induction (stacking $n$ blocks vs stacking $n + 1$ blocks), geometry, and intuitive physics.

**Most tasks should be solvable by humans.** To ensure that solving the tasks is theoretically possible, we manually solved most tasks using the same action space as the agent. We also provide scripts that allow researchers to explore the environments and attempt to solve tasks themselves.

**Tasks should range from very easy to extremely hard.** This is an important feature of Builder-Bench, meant to provide breadcrumbs of feedback to go from current algorithms capable of solving only the simplest tasks and agents that can build anything.

**Tasks should include some whose solutions are unknown even to the authors.** One aim of BuilderBench is also to see if artificial agents can come up with solutions to problems whose solutions are unknown. Hence, we have included a small minority of tasks which we were not able to solve.

## 6 TRAINING AND EVALUATION PROTOCOLS.

To evaluate open-ended exploration and generalization, we design the multi-task self-supervised protocol. Because this protocol contains various challenges, we also provide a debug single-task supervised protocol meant to provide additional feedback for researchers. We describe both protocols below and provide additional details in Appendix A.

**Multi-task self-supervised protocol.** The agent interacts with the environment, but does not receive any task specification during training. The agent's goal is to explore its environment to acquire general knowledge and skills that might help it to solve future tasks. The agent has to learn a task conditioned policy (Kaelbling, 1993), which can take as input a state ($\mathbb{R}^{11+13n}$) as well as a task specification ($\mathbb{R}^{3k}$). Each environment has a number of hand-designed tasks associated with it Appendix E. The agent is evaluated by running its task-conditioned policy on these tasks and measuring the reward obtained by it.

During training, it is highly unlikely that the agents will have seen these hand-designed tasks. Hence, to solve this protocol, agents will have to learn general reusable skills and concepts through purely self-supervised interaction. Many of these tasks are very difficult and unsolvable by the initial algorithms we tried. To provide additional feedback for algorithmic development, we also provide a simpler "training-wheels" protocol.

**Single-task supervised protocol.** In this standard RL protocol, the agents interact with a single environment to solve a single task from the task-suite. Each environment comes with a reward $r_t$. For each task, we currently support two types of reward functions – dense vs sparse, and permutation variant vs invariant to the cube order. By default, the rewards used are dense and permutation invariant to the cube order. Exact details of the reward functions are provided in Appendix A.2. The agent's objective is to learn a policy that maximizes the expected sum of rewards (Sutton & Barto, 2018).

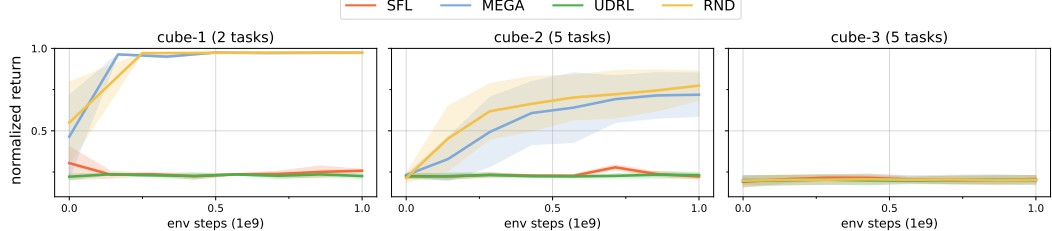

Figure 6: **Self-supervised evaluation on BuilderBench task-suite.** We evaluate MEGA (Pitis et al., 2020) and SFL (Rutherford et al., 2024) on 12 of the lowest complexity (yet still difficult) tasks from our task suite. The results show that directly using these algorithms out of the box only succeeds for the simplest tasks.

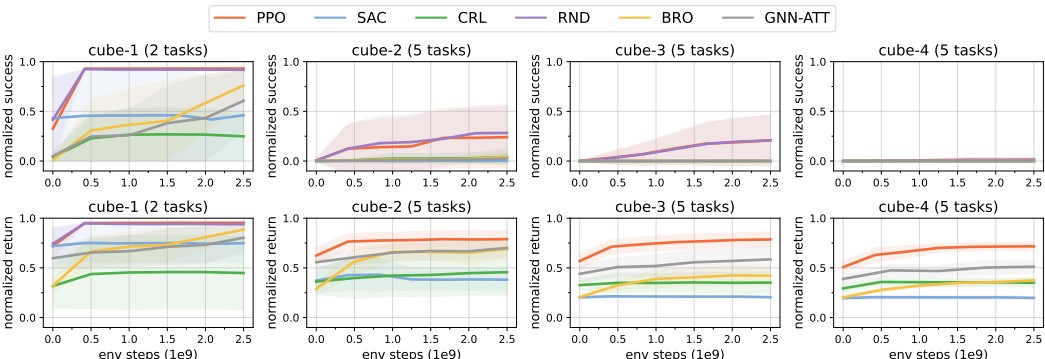

Figure 7: **Training on the test goals.** Training on the test goals improves both the returns and success achieved by the best agents. However as the number of cubes and the complexity of the tasks increase, current algorithms are not able to achieve a non zero success.

Although this setup does not directly evaluate generalization, it makes the problem of building general agents much more approachable. For instance, researchers could study various design choices or estimate whether an architecture is even capable of representing the solution to a complex task. Finally, because of the diversity of tasks, spanning a wide range of difficulties and reward formulations, this protocol is itself a useful benchmark for many RL fields such as goal conditioned, hierarchical and multi-task RL.

## 7 BENCHMARKING AND DISCUSSION.

In our experiments, we present benchmarking results for existing algorithms suited for the self-supervised as well as the supervised protocol. All experimental results are reported across three seeds. We also attempt to solve tasks using large language models. In Appendix B, we plot the training speed of various environments in BuilderBench.

**Multi-task self-supervised protocol.** We implemented four algorithms, sampling for learnability (**SFL** (Rutherford et al., 2024)), maximum entropy gain exploration (**MEGA** (Pitis et al., 2020)), upside down RL (**UDRL**) (Schmidhuber, 2020) and random network distillation (**RND**) (Burda et al., 2018). SFL and MEGA sample autotelic goals from previously visited states, for the agent to learn to reach them. SFL is an unsupervised environment design (Dennis et al., 2020) algorithm, which samples goals with the highest learnability (variance of success). MEGA is an unsupervised goal sampling (Florensa et al., 2018; OpenAI et al., 2021) algorithm, which samples goals inversely proportional to their visitation density. Both algorithms are implemented using proximal policy optimization (PPO) (Schulman et al., 2017).UDRL and RND are self-supervised algorithms. UDRL learns to reach previously explored goals using hindsight relabelling (Andrychowicz et al., 2018) and RND explores the environment using an intrinsic reward bonus. Both of these algorithms sample data collection goals using MEGA. All algorithms are trained in environments with one, two and three cubes and the learned policies are tested on the respective tasks from the task-suite Appendix E at various points during training. We report normalized episodic success and returns in Figure 6.

As seen in Figure 6, both algorithms achieve trivial performance on tasks with three cubes. MEGA is able to complete both tasks with one cube, and shows improvement on tasks with two cubes. While these results indicate that the tested algorithms are not directly scalable to complex tasks, it primarily underscores the inherent difficulty of the task setup itself. We believe that research in developing new algorithms (or revisiting old ones) is required to solve these tasks.

**Single-task supervised protocol.** For this protocol, we benchmark six RL algorithms, proximal policy optimization (**PPO**) (Schulman et al., 2017), soft actor critic (**SAC**) (Haarnoja et al., 2018), contrastive RL (**CRL**) (Eysenbach et al., 2022), random network distillation (**RND**) (Burda et al., 2019), bigger-regularized-optimistic (**BRO**) (Nauman et al., 2024) and graph-attention-network (**GNN-ATT**) (Ghasemipour et al., 2022). The benchmarking results on 17 tasks are provided in Figure 7. All experiments use dense rewards.

## 7.1 EVALUATING LARGE LANGUAGE MODELS

It has been shown that scaling pretraining and inference-time compute can significantly enhance the reasoning abilities of language models (Kaplan et al., 2020). To test whether the latest proprietary models can solve tasks from our task-suite, we evaluated ChatGPT-5[3] and Gemini 2.5 Pro (Team, 2025) on all five tasks discussed in Section 5.1. Each model was provided with a descriptive prompt about the environment and the task. The goal of the model was to provide a high-level open-loop plan in language, such that following this plan would stably build the target structure. A simple example task with a correct solution was also included in the prompt (see Appendix D for the exact prompts and solutions). Figure 8 shows that both models, despite using inference-time compute, are not able

Figure 8: **Evaluating language models on Builder-Bench.**

| Task Name | ChatGPT-5 | Gemini 2.5 Pro |
|---|---|---|
| T block | ✗ | ✗ |
| Four cube packing | ✗ | ✗ |
| Hexagonal Portal | ✗ | ✗ |
| Leaning tower | ✗ | ✗ |
| Maximum Overhang | ✗ | ✗ |

to provide the correct high-level plan to solve any of the tasks. While this is not meant to be an extensive evaluation of current models' abilities, it highlights how solving our tasks requires non-obvious steps of reasoning that are beyond what current models can achieve through scaling alone.

## 8 LIMITATIONS AND CONCLUSION

Although BuilderBench contains tasks that require diverse and open-ended skills, it does not focus on evaluating other problems like stochasticity, partial observability, or multi-agent learning. However, we expect extending BuilderBench to incorporate these settings should be easy. Another limitation is that we have not provided implementations for all approaches for open-ended exploration which exist in the literature (see discussion in Section 2). This is outside the scope of the paper. The main aim of the paper is to present an effective benchmark to accelerate research on scalable and generalizable learning through open-ended exploration.

Developing agents that can learn through open-ended exploration and generalize across diverse tasks remains an open problem in AI. Current AI models are pretrained on human generated data. As a result, they largely lack the ability to explore and learn through interaction. We have designed BuilderBench, to accelerate research towards agents that learn to explore in an open-ended environment and generalize to diverse tasks. Tasks in BuilderBench are designed to elicit long-horizon planning and reasoning abilities, many implicitly requiring agents to solve problems in physics and mathematics. BuilderBench provides a common framework for studying problems like open-ended exploration, generalization and embodied reasoning. We expect that the resulting research will advance the development of agents that solve problems by interacting with the real world.

---

[3]https://openai.com/index/introducing-gpt-5/

## 9 REPRODUCIBILITY STATEMENT

All experiments in this paper are completely reproducible. We have attached our code for the simulator, the task-suite and the implementation of all algorithms as a part of the supplemental materials. Additionally, BuilderBench is based on MuJoCo (Todorov et al., 2012) and Jax (Bradbury et al., 2018) both of which are open-sourced libraries. For experiments using proprietary language models, we have provided the exact models Section 7.1 and the prompts Appendix D which were used for experiments.

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

# A  ENVIRONMENT DETAILS

## A.1  EPISODE LENGTHS

The episode length depends on the number of cubes present in the environment ($N$). For supervised tasks, the episode length is $100 + 100 \times N$ and for self-supervised tasks, the episode length is $500 \times N$.

## A.2  REWARD FUNCTIONS

There are two types of reward functions provided in the benchmark, sparse and dense. The sparse reward is equal to -1 for all timesteps where the cubes do not form the target structure and 0 if the cubes form the target structure. Cubes are said to form the target structure if the distance between each cube and its corresponding target is less than 2 centimeters. At each timestep, every cube is assigned a specific target position from the target structure. This assignment is calculated such that the total sum of distances between the cubes and their assigned targets is minimized. This is a convex optimization problem and can be solved efficiently with GPUs using the hungarian algorithm implemented in jax Bradbury et al. (2018). The sparse rewards are calculated by checking whether the best assigned distances are all less than 2 centimeters. The dense rewards are calculated by applying $1 - \tanh(x)$ to the best assigned distances and summing them over all cubes. As distances tend to zero, the reward tends to N (number of cubes in the environment).

The permutation sensitive variants of both the sparse and dense rewards are calculated similarly, but without solving the assignment problem. Each cube is already assigned a specific target position by the task, and this assignment is used to estimate both the dense and the sparse rewards.

# B  TRAINING SPEED.

In Figure 9, we plot the average environment steps per second for PPO (Figure 7) and MEGA (Figure 6). The PPO experiments were run on a single A100 GPU with 12 CPU threads and the MEGA experiments were run on a single A6000 GPU with 12 CPU threads. The speed of the training depends on many factors like number of parallel environments, the update to data ratio, type of GPU and the number of CPU threads. The speed also depends on the environment step-time, number of cubes and the type of interactions in the environments (interactions with more active contacts typically take longer time to execute). An example of this can be seen in the right

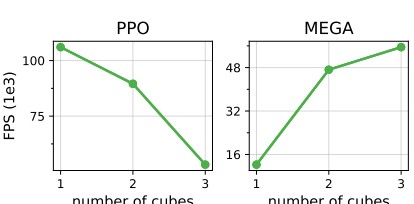

Figure 9: **Environment steps per second during PPO and MEGA training runs**

plot in Figure 9. For the self-supervised protocol, the environment step-time is set to be small for better physics approximation. MEGA with one block learns to pick the block up and move it around which we found to be inversely correlated with speed. The speeds could be scaled with more CPU threads, parallel environments, and faster algorithms for physics approximation.

Although not precise, comparing these speeds with fig. 2 from Matthews et al. (2024) shows that BuilderBench is between $10^2 - 10^3$ times faster than Minecraft (Guss et al., 2019), $2 - 20$ faster than Nethack and Procgen (Küttler et al., 2020; Cobbe et al., 2020), and $10 - 10^2$ times faster than Crafter (Hafner, 2022).

Table 1: Qualitative Comparison of BuilerBench with previous benchmarks

| Benchmark | Extensive reasoning task-suite | Hardware accelerated | Primary Focus |
|---|:---:|:---:|---|
| BuilerBench (ours) | ✓ | ✓ | 1. Self-supervised exploration
2. Zero shot reasoning
3. Learning from scratch |
| ManiSkill (Mu et al., 2021) | ✗ | ✓ | 1. Robot learning
2. Fast simulation and training for manipulation tasks. |
| FurnitureBench (Heo et al., 2023) | ✗ | ✗ | 1. Long-horizon assembly
2. Imitation learning and Offline RL |
| BabyAI (MCB et al., 2019) | ✗ | ✗ | 1. Human in the loop learning
2. Language grounded reasoning |
| Kinetix (Matthews et al., 2025) | ✗ | ✓ | 1. Zero shot reasoning
2. Unsupervised environment design |
| Craftium (Malagón et al., 2025) | ✗ | ✗ | 1. LLM based agents
2. Code based environment generation |
| MiniGrid (MCB et al., 2023) | ✗ | ✗ | 1. Exploration
2. Curriculum Learning |

## C  QUALITATIVE COMPARISON WITH PRIOR WORK.

In this section, we provide a qualitative comparison of BuilderBench with prior work. Table 1 compares the primary focus of BuilderBench with various benchmarks, *highlighting the need for a new benchmark that focuses on learning to reason via exploration and trial and error from scratch.*

We argue that the components of BuilderBench are necessary to facilitate research on this problem:

**Tasks which demonstrably require reasoning.** We have provided an extensive task suite which require skills such as logical reasoning (commutativity and associativity of pick and place ordering), geometrical reasoning (maximizing overhangs, packing problems) and intuitive physics (gravity, friction, toppling, balancing). Tasks also require reasoning about counterweights, buttresses, and temporary scaffolding. Designing such tasks is not trivial. In comparison, benchmarks like ManiSkill (Mu et al., 2021) focus on simpler tasks which require pick and place primitives.

**Scaling interaction speed.** Training agents to solve such complex problems will presumably require large amounts of interaction data. While there are complex environments like MineCraft (Guss et al., 2019; Malagón et al., 2025) that are also centered around building blocks, the simulators for these environments are about $10^2 - 10^3$ times slower that BuilderBench.

**Focus on training from trial and error.** BuilderBench focuses on agents that learn purely from trial and error whereas benchmarks like FurnitureBench (Heo et al., 2023) focus on solve long-horizon assembly using demonstrations and offline data.

# D   EVALUATING LANGUAGE MODELS: PROMPTS AND SOLUTIONS

## D.1   T BLOCK

---

**User**

You are an agent that controls a two-finger robotic gripper (Robotiq 2F-85) which can navigate a constrained 3D space using position actuators for controlling x,y,z directions and yaw. The robot hand has two fingers with maximum width equal to 0.085 meters. The environment consists of a variable number of cubes of size 0.04 meters. The environment is implemented in MuJoCo and approximates Newtonian physics. All length position coordinates have units in meters and the yaw will be measured in radians.

Problem description - You will be given the positions and the orientations of all the N cubes in the environment. You will also be given the target center positions of K cubes, where K can be less or equal to N. The target center positions correspond to a target structure that you need to build such that it always remains stable. The structures which are in unstable equilibria, that would fall apart after applying minimal perturbation, are not considered stable. Your goal is to provide high level language instructions for the gripper to follow such that it can build the stable target structure. All steps should be physically possible and implementing those steps one after the other should stably result in building the target structure. Only provide a list of an open loop plan consisting of simple instructions, for example, pick cube 1, place cube 1 at 0.1,0,0.02 with yaw 0, rotate cube 2 by pi/3 radians, nudge the cube at [0.1,0,0.02] to [0.12,0,0.02], etc. You can think before providing the final answer but please separate the final answer by saying "Final answer:" before providing the ordered plan of high level instructions.

The following is an example of a simple task followed by its solution:

Task:
N = 1
K = 1
Start location = [0.05, 0.0, 0.02]
Target location = [0.1, 0.0, 0.02]

Final answer:
1) Pick up the cube from [0.05, 0.0, 0.02].
2) Place the cube at [0.1, 0.0, 0.02] with a yaw of 0 radians.

Here is the task:
N = 3
K = 3
Start location = [ [ 0.05, -0.08, 0.02], [ 0.05, 0. , 0.02], [ 0.05, 0.08, 0.02] ]
Target location = [ [ 0.1, 0.02, 0.06], [ 0.1 , -0.02, 0.06], [ 0.1 , 0. , 0.02] ]

---

**ChatGPT-5**

Pick up the cube from [0.05, 0.0, 0.02].
Place the cube at [0.1, 0.0, 0.02] with a yaw of 0 radians.
Pick up the cube from [0.05, -0.08, 0.02].
Place the cube at [0.1, -0.02, 0.06] with a yaw of 0 radians.
Pick up the cube from [0.05, 0.08, 0.02].
Place the cube at [0.1, 0.02, 0.06] with a yaw of 0 radians.

**Gemini 2.5 Pro**

Pick up the cube from [0.05, 0.0, 0.02].
Place the cube at [0.1, 0.0, 0.02] with a yaw of 0 radians.
Pick up the cube from [0.05, -0.08, 0.02].
Place the cube at [0.1, -0.02, 0.06] with a yaw of 0 radians.
Pick up the cube from [0.05, 0.08, 0.02].
Place the cube at [0.1, 0.02, 0.06] with a yaw of 0 radians.

## D.2 FOUR CUBE PACKING

---

**User**

You are an agent that controls a two-finger robotic gripper (Robotiq 2F-85) which can navigate a constrained 3D space using position actuators for controlling x,y,z directions and yaw. The robot hand has two fingers with maximum width equal to 0.085 meters. The environment consists of a variable number of cubes of size 0.04 meters. The environment is implemented in MuJoCo and approximates Newtonian physics. All length position coordinates have units in meters and the yaw will be measured in radians.

Problem description - You will be given the positions and the orientations of all the N cubes in the environment. You will also be given the target center positions of K cubes, where K can be less or equal to N. The target center positions correspond to a target structure that you need to build such that it always remains stable. The structures which are in unstable equilibria, that would fall apart after applying minimal perturbation, are not considered stable. Your goal is to provide high level language instructions for the gripper to follow such that it can build the stable target structure. All steps should be physically possible and implementing those steps one after the other should stably result in building the target structure. Only provide a list of an open loop plan consisting of simple instructions, for example, pick cube 1, place cube 1 at 0.1,0,0.02 with yaw 0, rotate cube 2 by pi/3 radians, nudge the cube at [0.1,0,0.02] to [0.12,0,0.02], etc. You can think before providing the final answer but please separate the final answer by saying "Final answer:" before providing the ordered plan of high level instructions.

The following is an example of a simple task followed by its solution:

Task:
N = 1
K = 1
Start location = [0.05, 0.0, 0.02]
Target location = [0.1, 0.0, 0.02]

Final answer:
1) Pick up the cube from [0.05, 0.0, 0.02].
2) Place the cube at [0.1, 0.0, 0.02] with a yaw of 0 radians.

Here is the task:
N = 4
K = 4
Start location = [ [0.05, -0.12, 0.02], [0.05, -0.04, 0.02], [0.05, 0.04, 0.02], [0.05, 0.12, 0.02] ]
Target location = [ [0.1, 0.02828427, 0.02], [0.1, -0.02828427, 0.02], [0.12828427, 0.0, 0.02], [0.07171573, 0.0, 0.02] ]

---

**ChatGPT-5**

Pick up the cube from [0.05, -0.12, 0.02].
Place the cube at [0.1, -0.02828427, 0.02] with a yaw of 0 radians.
Pick up the cube from [0.05, 0.12, 0.02].
Place the cube at [0.1, 0.02828427, 0.02] with a yaw of 0 radians.
Pick up the cube from [0.05, -0.04, 0.02].
Place the cube at [0.07171573, 0.0, 0.02] with a yaw of 0 radians.
Pick up the cube from [0.05, 0.04, 0.02].
Place the cube at [0.12828427, 0.0, 0.02] with a yaw of 0 radians.

**Gemini 2.5 Pro**

Pick up the cube from [0.05, -0.12, 0.02].
Place the cube at [0.1, 0.02828427, 0.02] with a yaw of 0 radians.
Pick up the cube from [0.05, -0.04, 0.02].
Place the cube at [0.1, -0.02828427, 0.02] with a yaw of 0 radians.
Pick up the cube from [0.05, 0.04, 0.02].
Place the cube at [0.12828427, 0.0, 0.02] with a yaw of 0 radians.
Pick up the cube from [0.05, 0.12, 0.02].
Place the cube at [0.07171573, 0.0, 0.02] with a yaw of 0 radians.

## D.3 Hexagonal Portal

---

**User**

You are an agent that controls a two-finger robotic gripper (Robotiq 2F-85) which can navigate a constrained 3D space using position actuators for controlling x,y,z directions and yaw. The robot hand has two fingers with maximum width equal to 0.085 meters. The environment consists of a variable number of cubes of size 0.04 meters. The environment is implemented in MuJoCo and approximates Newtonian physics. All length position coordinates have units in meters and the yaw will be measured in radians.

Problem description - You will be given the positions and the orientations of all the N cubes in the environment. You will also be given the target center positions of K cubes, where K can be less or equal to N. The target center positions correspond to a target structure that you need to build such that it always remains stable. The structures which are in unstable equilibria, that would fall apart after applying minimal perturbation, are not considered stable. Your goal is to provide high level language instructions for the gripper to follow such that it can build the stable target structure. All steps should be physically possible and implementing those steps one after the other should stably result in building the target structure. Only provide a list of an open loop plan consisting of simple instructions, for example, pick cube 1, place cube 1 at 0.1,0,0.02 with yaw 0, rotate cube 2 by pi/3 radians, nudge the cube at [0.1,0,0.02] to [0.12,0,0.02], etc. You can think before providing the final answer but please separate the final answer by saying "Final answer:" before providing the ordered plan of high level instructions.

The following is an example of a simple task followed by its solution:

Task:
N = 1
K = 1
Start location = [0.05, 0.0, 0.02]
Target location = [0.1, 0.0, 0.02]

Final answer:
1) Pick up the cube from [0.05, 0.0, 0.02].
2) Place the cube at [0.1, 0.0, 0.02] with a yaw of 0 radians.

Here is the task:
N = 8
K = 8
Start location = [ [0.05, -0.24, 0.02], [0.05, -0.18, 0.02], [0.05, -0.12, 0.02], [0.05, -0.04, 0.02], [0.05, 0.04, 0.02], [0.05, 0.12, 0.02], [0.05, 0.18, 0.02], [0.05, 0.24, 0.02] ]
Target location = [ [0.1, 0.02, 0.02], [0.1, -0.02, 0.02], [0.1, 0.04, 0.06], [0.1, -0.04, 0.06], [0.1, 0.02, 0.1], [0.1, -0.02, 0.1], [0.1, 0.1, 0.02], [0.1, -0.1, 0.02] ]

---

**ChatGPT-5**

Pick up the cube from [0.05, 0.18, 0.02]. Place the cube at [0.1, 0.02, 0.02] with a yaw of 0 radians.
Pick up the cube from [0.05, 0.24, 0.02]. Place the cube at [0.1, 0.1, 0.02] with a yaw of 0 radians.
Pick up the cube from [0.05, 0.12, 0.02]. Place the cube at [0.1, 0.04, 0.06] with a yaw of 0 radians.
Pick up the cube from [0.05, 0.04, 0.02]. Place the cube at [0.1, 0.02, 0.1] with a yaw of 0 radians.
Pick up the cube from [0.05, -0.18, 0.02]. Place the cube at [0.1, -0.02, 0.02] with a yaw of 0 radians.

Pick up the cube from [0.05, -0.24, 0.02]. Place the cube at [0.1, -0.1, 0.02] with a yaw of 0 radians.
Pick up the cube from [0.05, -0.12, 0.02]. Place the cube at [0.1, -0.04, 0.06] with a yaw of 0 radians.
Pick up the cube from [0.05, -0.04, 0.02]. Place the cube at [0.1, -0.02, 0.1] with a yaw of 0 radians.

**Gemini 2.5 Pro**

Pick up the cube from [0.05, -0.24, 0.02].
Place the cube at [0.1, 0.02, 0.02] with a yaw of 0 radians.
Pick up the cube from [0.05, -0.18, 0.02].
Place the cube at [0.1, -0.02, 0.02] with a yaw of 0 radians.
Pick up the cube from [0.05, -0.12, 0.02].
Place the cube at [0.1, 0.1, 0.02] with a yaw of 0 radians.
Pick up the cube from [0.05, -0.04, 0.02].
Place the cube at [0.1, -0.1, 0.02] with a yaw of 0 radians.
Pick up the cube from [0.05, 0.04, 0.02].
Place the cube at [0.1, 0.04, 0.06] with a yaw of 0 radians.
Pick up the cube from [0.05, 0.12, 0.02].
Place the cube at [0.1, -0.04, 0.06] with a yaw of 0 radians.
Pick up the cube from [0.05, 0.18, 0.02].
Place the cube at [0.1, 0.02, 0.1] with a yaw of 0 radians.
Pick up the cube from [0.05, 0.24, 0.02].
Place the cube at [0.1, -0.02, 0.1] with a yaw of 0 radians.

## D.4 LEANING TOWER

> **User**
>
> You are an agent that controls a two-finger robotic gripper (Robotiq 2F-85) which can navigate a constrained 3D space using position actuators for controlling x,y,z directions and yaw. The robot hand has two fingers with maximum width equal to 0.085 meters. The environment consists of a variable number of cubes of size 0.04 meters. The environment is implemented in MuJoCo and approximates Newtonian physics. All length position coordinates have units in meters and the yaw will be measured in radians.
>
> Problem description - You will be given the positions and the orientations of all the N cubes in the environment. You will also be given the target center positions of K cubes, where K can be less or equal to N. The target center positions correspond to a target structure that you need to build such that it always remains stable. The structures which are in unstable equilibria, that would fall apart after applying minimal perturbation, are not considered stable. Your goal is to provide high level language instructions for the gripper to follow such that it can build the stable target structure. All steps should be physically possible and implementing those steps one after the other should stably result in building the target structure. Only provide a list of an open loop plan consisting of simple instructions, for example, pick cube 1, place cube 1 at 0.1,0,0.02 with yaw 0, rotate cube 2 by pi/3 radians, nudge the cube at [0.1,0,0.02] to [0.12,0,0.02], etc. You can think before providing the final answer but please separate the final answer by saying "Final answer:" before providing the ordered plan of high level instructions.
>
> The following is an example of a simple task followed by its solution:
>
> Task:
> N = 1
> K = 1
> Start location = [0.05, 0.0, 0.02]
> Target location = [0.1, 0.0, 0.02]
>
> Final answer:
> 1) Pick up the cube from [0.05, 0.0, 0.02].
> 2) Place the cube at [0.1, 0.0, 0.02] with a yaw of 0 radians.
>
> Here is the task:
> N = 9
> K = 9
> Start location = [ [0.05, -0.3, 0.02], [0.05, -0.24, 0.02], [0.05, -0.16, 0.02], [0.05, -0.08, 0.02], [0.05, 0.0, 0.02], [0.05, 0.08, 0.02], [0.05, 0.16, 0.02], [0.05, 0.24, 0.02], [0.05, 0.3, 0.02] ]
> Target location = [ [0.1, 0.0, 0.02], [0.1, -0.04, 0.02], [0.1, 0.02, 0.06], [0.1, -0.02, 0.06], [0.1, 0.04, 0.1], [0.1, 0.0, 0.1], [0.1, 0.01, 0.14], [0.1, 0.12, 0.02], [0.1, 0.16, 0.02] ]

> **ChatGPT-5**
>
> Pick up the cube from [0.05, -0.3, 0.02]. Place the cube at [0.1, -0.04, 0.02] with a yaw of 0 radians.
> Pick up the cube from [0.05, -0.24, 0.02]. Place the cube at [0.1, 0.0, 0.02] with a yaw of 0 radians.
> Pick up the cube from [0.05, 0.24, 0.02]. Place the cube at [0.1, 0.12, 0.02] with a yaw of 0 radians.
> Pick up the cube from [0.05, 0.3, 0.02]. Place the cube at [0.1, 0.16, 0.02] with a yaw of 0 radians.
> Pick up the cube from [0.05, -0.16, 0.02]. Place the cube at [0.1, -0.02, 0.06] with a yaw of 0 radians.

Pick up the cube from [0.05, -0.08, 0.02]. Place the cube at [0.1, 0.02, 0.06] with a yaw of 0 radians.
Pick up the cube from [0.05, 0.0, 0.02]. Place the cube at [0.1, 0.0, 0.10] with a yaw of 0 radians.
Pick up the cube from [0.05, 0.08, 0.02]. Place the cube at [0.1, 0.04, 0.10] with a yaw of 0 radians.
Pick up the cube from [0.05, 0.16, 0.02]. Place the cube at [0.1, 0.01, 0.14] with a yaw of 0 radians.

**Gemini 2.5 Pro**

Pick up the cube from [0.05, -0.3, 0.02] and place it at [0.1, -0.04, 0.02] with a yaw of 0 radians.
Pick up the cube from [0.05, -0.24, 0.02] and place it at [0.1, 0.0, 0.02] with a yaw of 0 radians.
Pick up the cube from [0.05, -0.16, 0.02] and place it at [0.1, 0.12, 0.02] with a yaw of 0 radians.
Pick up the cube from [0.05, -0.08, 0.02] and place it at [0.1, 0.16, 0.02] with a yaw of 0 radians.
Pick up the cube from [0.05, 0.0, 0.02] and place it at [0.1, -0.02, 0.06] with a yaw of 0 radians.
Pick up the cube from [0.05, 0.08, 0.02] and place it at [0.1, 0.02, 0.06] with a yaw of 0 radians.
Pick up the cube from [0.05, 0.16, 0.02] and place it at [0.1, 0.0, 0.1] with a yaw of 0 radians.
Pick up the cube from [0.05, 0.24, 0.02] and place it at [0.1, 0.04, 0.1] with a yaw of 0 radians.
Pick up the cube from [0.05, 0.3, 0.02] and place it at [0.1, 0.01, 0.14] with a yaw of 0 radians.

## D.5 Maximum Overhang

---

**User**

You are an agent that controls a two-finger robotic gripper (Robotiq 2F-85) which can navigate a constrained 3D space using position actuators for controlling x,y,z directions and yaw. The robot hand has two fingers with maximum width equal to 0.085 meters. The environment consists of a variable number of cubes of size 0.04 meters. The environment is implemented in MuJoCo and approximates Newtonian physics. All length position coordinates have units in meters and the yaw will be measured in radians.

Problem description - You will be given the positions and the orientations of all the N cubes in the environment. You will also be given the target center positions of K cubes, where K can be less or equal to N. The target center positions correspond to a target structure that you need to build such that it always remains stable. The structures which are in unstable equilibria, that would fall apart after applying minimal perturbation, are not considered stable. Your goal is to provide high level language instructions for the gripper to follow such that it can build the stable target structure. All steps should be physically possible and implementing those steps one after the other should stably result in building the target structure. Only provide a list of an open loop plan consisting of simple instructions, for example, pick cube 1, place cube 1 at 0.1,0,0.02 with yaw 0, rotate cube 2 by pi/3 radians, nudge the cube at [0.1,0,0.02] to [0.12,0,0.02], etc. You can think before providing the final answer but please separate the final answer by saying "Final answer:" before providing the ordered plan of high level instructions.

The following is an example of a simple task followed by its solution:

Task:
N = 1
K = 1
Start location = [0.05, 0.0, 0.02]
Target location = [0.1, 0.0, 0.02]

Final answer:
1) Pick up the cube from [0.05, 0.0, 0.02].
2) Place the cube at [0.1, 0.0, 0.02] with a yaw of 0 radians.

Here is the task:
N = 5
K = 3
Start location = [ [0.05, -0.16, 0.02], [0.05, -0.08, 0.02], [0.05, 0.0, 0.02], [0.05, 0.08, 0.02], [0.05, 0.16, 0.02] ]
Target location = [ [0.1, 0.0, 0.02], [0.1, 0.031, 0.14], [0.1, 0.16, 0.14] ]

---

**ChatGPT-5**

Pick up the cube from [0.05, 0.0, 0.02]. Place the cube at [0.1, 0.0, 0.02] with a yaw of 0 radians.
Pick up the cube from [0.05, -0.08, 0.02]. Place the cube at [0.1, 0.031, 0.02] with a yaw of 0 radians. (bottom of the pillar)
Pick up the cube from [0.05, 0.08, 0.02]. Place the cube at [0.1, 0.031, 0.06] with a yaw of 0 radians. (middle of the pillar)
Pick up the cube from [0.05, -0.16, 0.02]. Place the cube at [0.1, 0.031, 0.10] with a yaw of 0 radians. (top of the pillar)
Pick up the cube from [0.05, 0.16, 0.02]. Place the cube at [0.1, 0.031, 0.14] with a yaw of 0 radians. (target at z=0.14 realized)

> **Gemini 2.5 Pro**
>
> It is not possible to generate a plan for this task. The target structure is physically unstable.

# E  TASKS AND CHALLENGES IN BUILDERBENCH

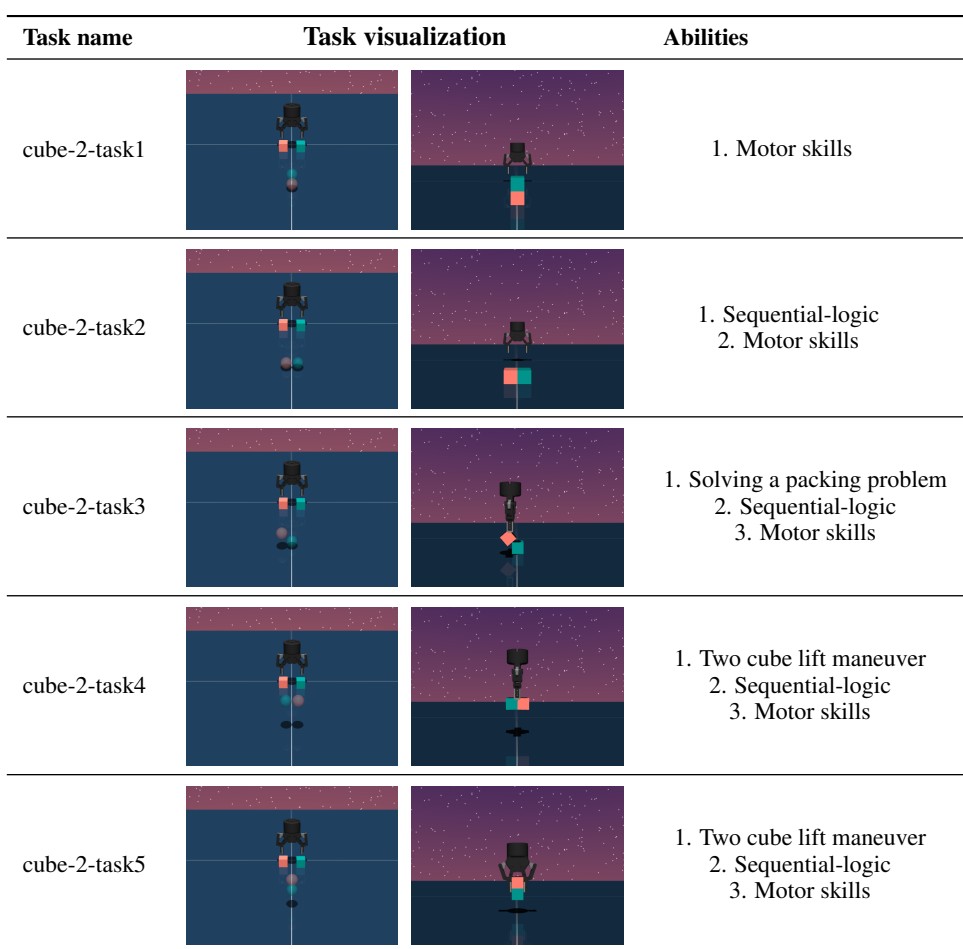

| Task name | Task visualization | | Abilities |
|---|---|---|---|
| cube-1-task1 | | | 1. Pick and Place |
| cube-1-task2 | | | 1. Pick and hold |

Table 2: **Tasks with 1 cube**.

| Task name | Task visualization | | Abilities |
|---|---|---|---|
| cube-2-task1 | | | 1. Motor skills |
| cube-2-task2 | | | 1. Sequential-logic
2. Motor skills |
| cube-2-task3 | | | 1. Solving a packing problem
2. Sequential-logic
3. Motor skills |
| cube-2-task4 | | | 1. Two cube lift maneuver
2. Sequential-logic
3. Motor skills |
| cube-2-task5 | | | 1. Two cube lift maneuver
2. Sequential-logic
3. Motor skills |

Table 3: **Tasks with 2 cubes**.

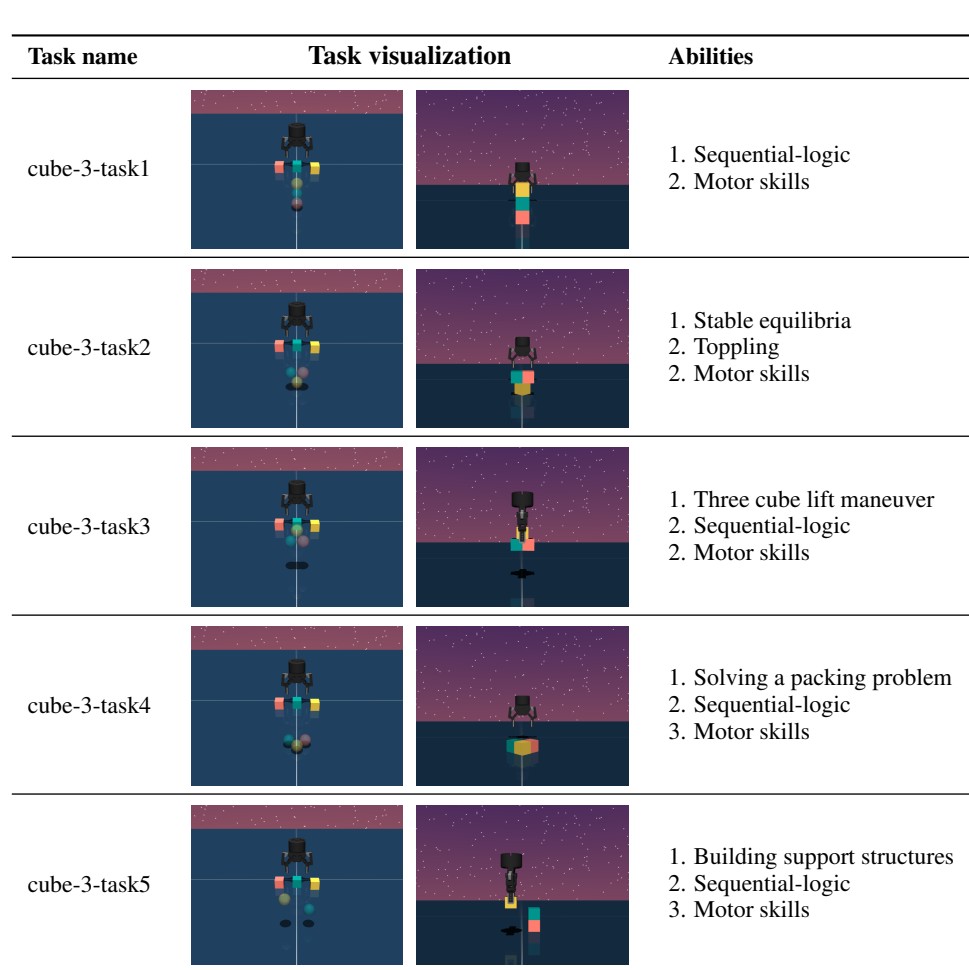

| Task name | Task visualization | | Abilities |
|-----------|-------------------|---|-----------|
| cube-3-task1 | | | 1. Sequential-logic
2. Motor skills |
| cube-3-task2 | | | 1. Stable equilibria
2. Toppling
2. Motor skills |
| cube-3-task3 | | | 1. Three cube lift maneuver
2. Sequential-logic
2. Motor skills |
| cube-3-task4 | | | 1. Solving a packing problem
2. Sequential-logic
3. Motor skills |
| cube-3-task5 | | | 1. Building support structures
2. Sequential-logic
3. Motor skills |

Table 4: **Tasks with 3 cubes**.

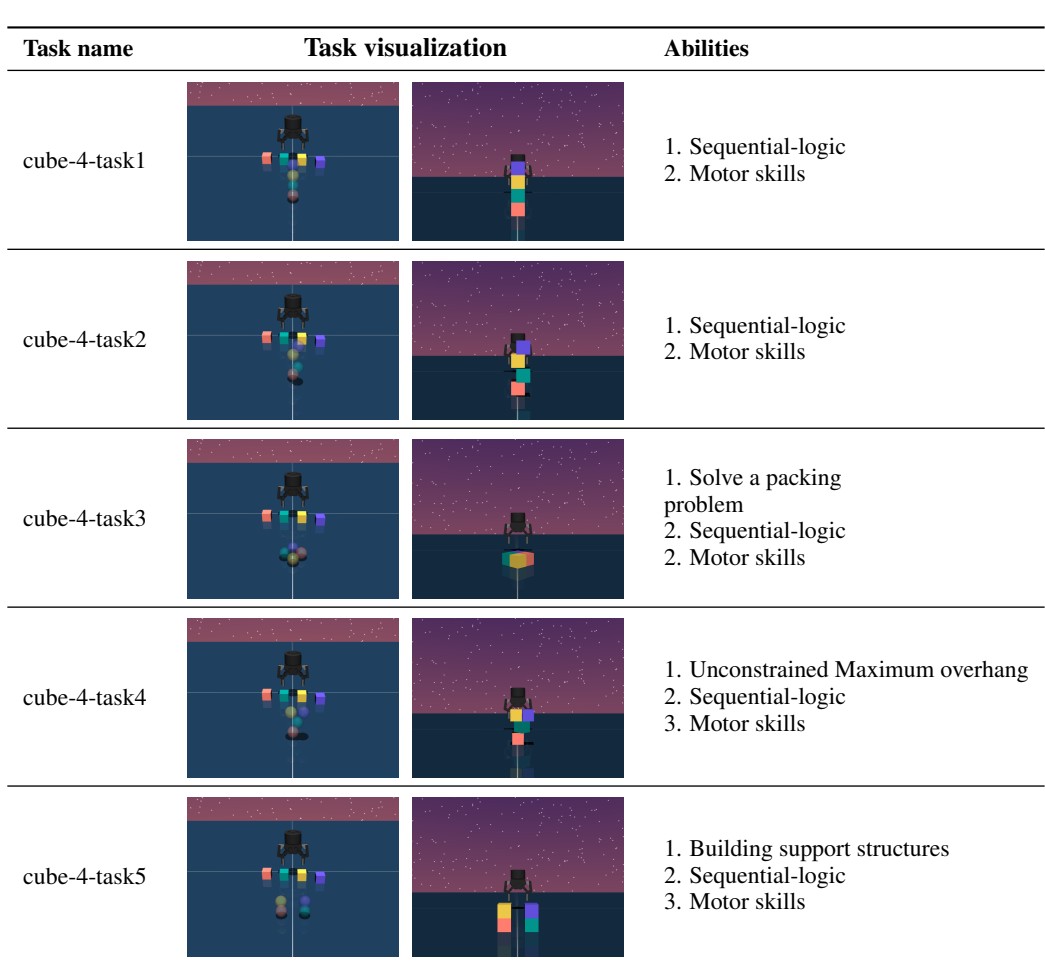

Table 5: **Tasks with 4 cubes**.

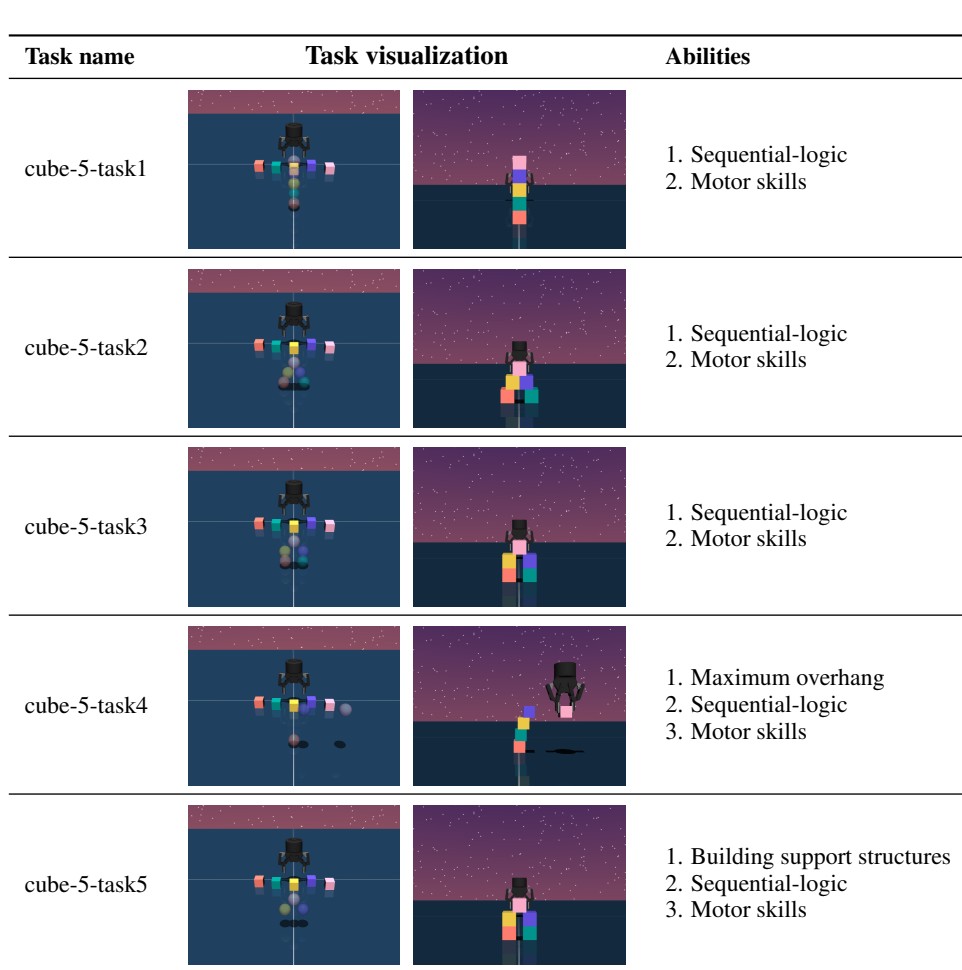

| Task name | Task visualization | | Abilities |
|-----------|-------------------|---|-----------|
| cube-5-task1 | | | 1. Sequential-logic
2. Motor skills |
| cube-5-task2 | | | 1. Sequential-logic
2. Motor skills |
| cube-5-task3 | | | 1. Sequential-logic
2. Motor skills |
| cube-5-task4 | | | 1. Maximum overhang
2. Sequential-logic
3. Motor skills |
| cube-5-task5 | | | 1. Building support structures
2. Sequential-logic
3. Motor skills |

Table 6: **Tasks with 5 cubes**.

| Task name | Task visualization | | Abilities |
|-----------|-------------------|---|-----------|
| cube-6-task1 | 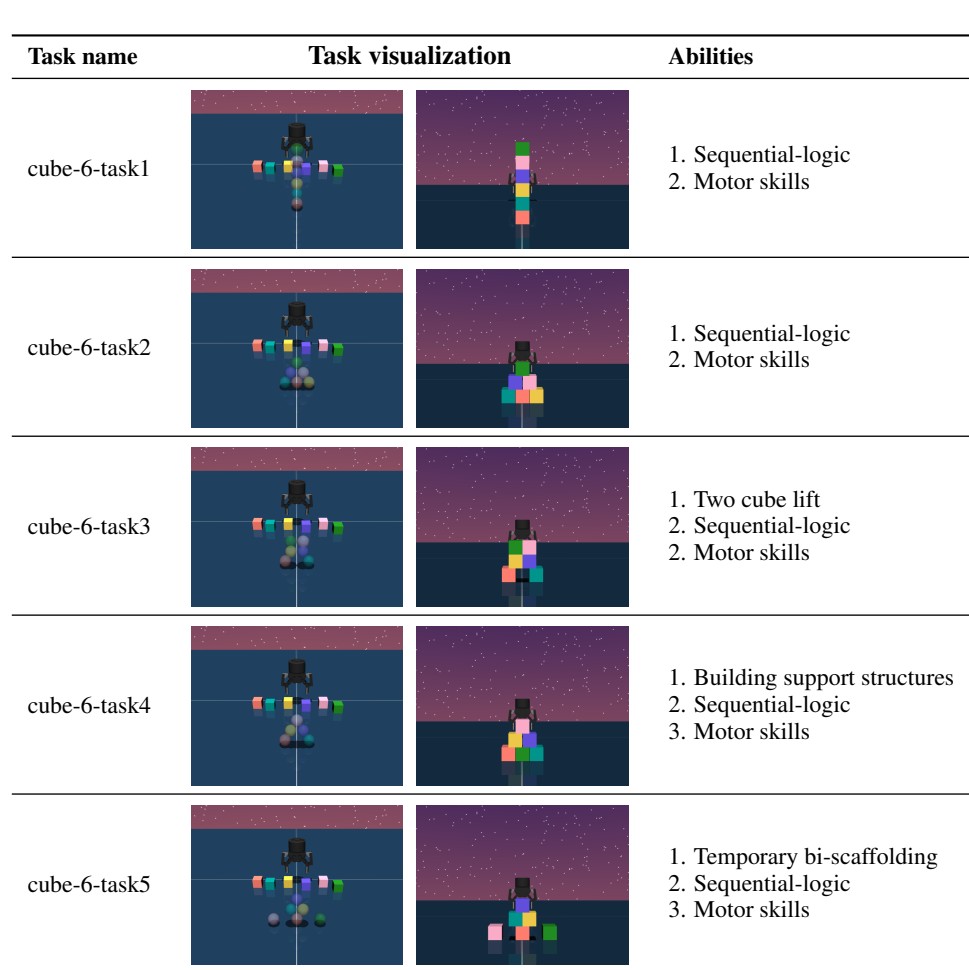 | | 1. Sequential-logic
2. Motor skills |
| cube-6-task2 | | | 1. Sequential-logic
2. Motor skills |
| cube-6-task3 | | | 1. Two cube lift
2. Sequential-logic
2. Motor skills |
| cube-6-task4 | | | 1. Building support structures
2. Sequential-logic
3. Motor skills |
| cube-6-task5 | | | 1. Temporary bi-scaffolding
2. Sequential-logic
3. Motor skills |

Table 7: **Tasks with 6 cubes**.

| Task name | Task visualization | | Abilities |
|-----------|-------------------|---|-----------|
| cube-7-task1 | 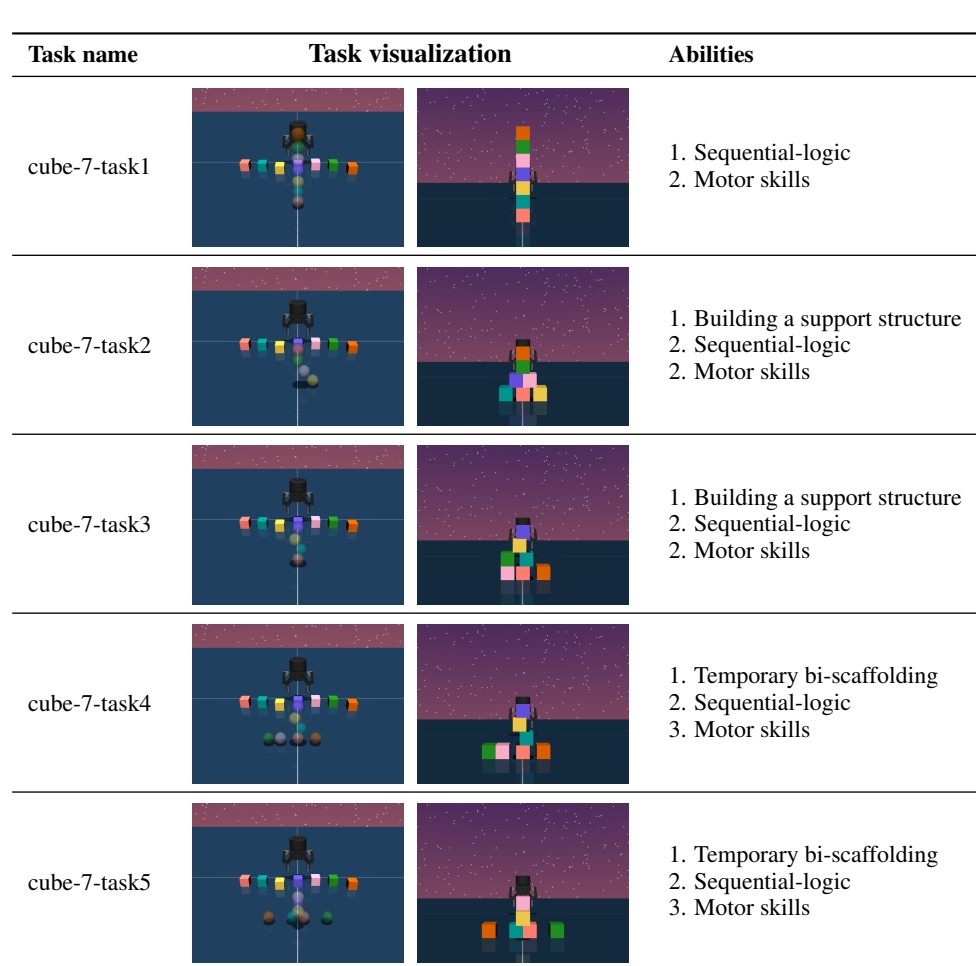 | | 1. Sequential-logic 
 2. Motor skills |
| cube-7-task2 | | | 1. Building a support structure 
 2. Sequential-logic 
 2. Motor skills |
| cube-7-task3 | | | 1. Building a support structure 
 2. Sequential-logic 
 2. Motor skills |
| cube-7-task4 | | | 1. Temporary bi-scaffolding 
 2. Sequential-logic 
 3. Motor skills |
| cube-7-task5 | | | 1. Temporary bi-scaffolding 
 2. Sequential-logic 
 3. Motor skills |

Table 8: **Tasks with 7 cubes**.

| Task name | Task visualization | Abilities |
|---|---|---|
| cube-8-task1 | 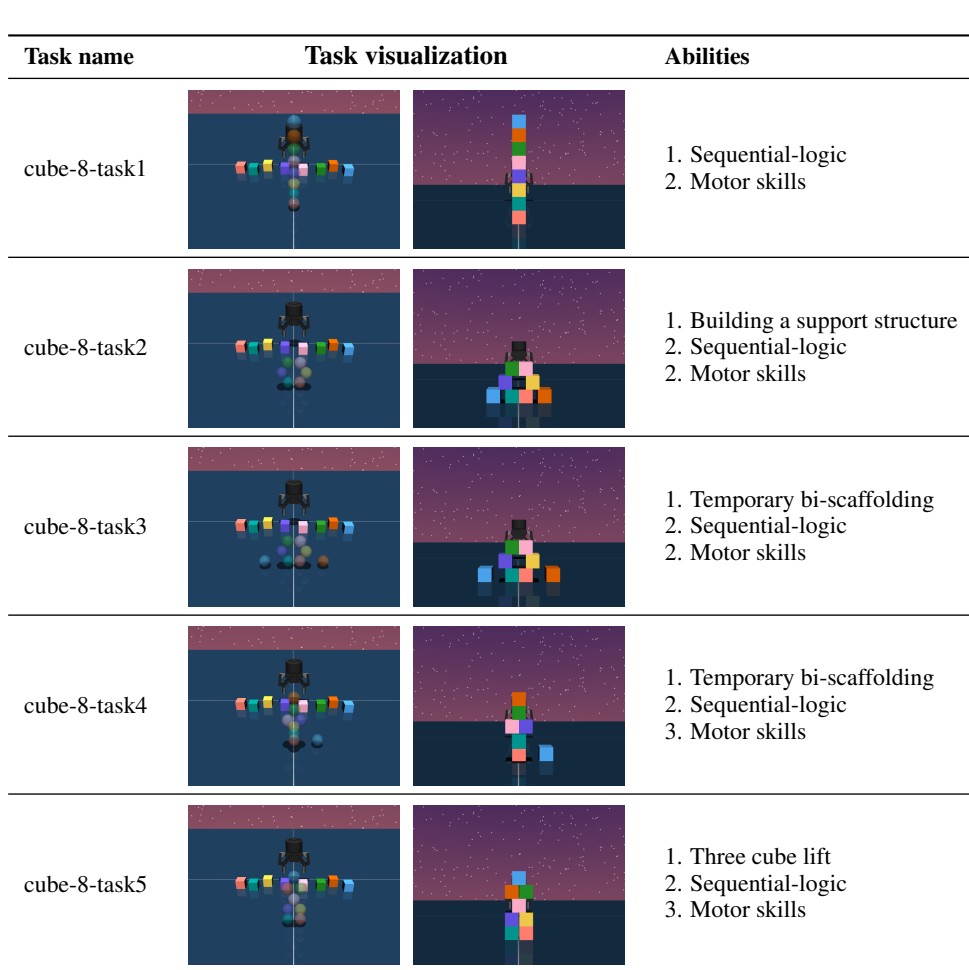 | 1. Sequential-logic
2. Motor skills |
| cube-8-task2 | | 1. Building a support structure
2. Sequential-logic
2. Motor skills |
| cube-8-task3 | | 1. Temporary bi-scaffolding
2. Sequential-logic
2. Motor skills |
| cube-8-task4 | | 1. Temporary bi-scaffolding
2. Sequential-logic
3. Motor skills |
| cube-8-task5 | | 1. Three cube lift
2. Sequential-logic
3. Motor skills |

Table 9: **Tasks with 8 cubes**.

| Task name | Task visualization | | Abilities |
|-----------|--------------------|--|-----------|
| cube-9-task1 | 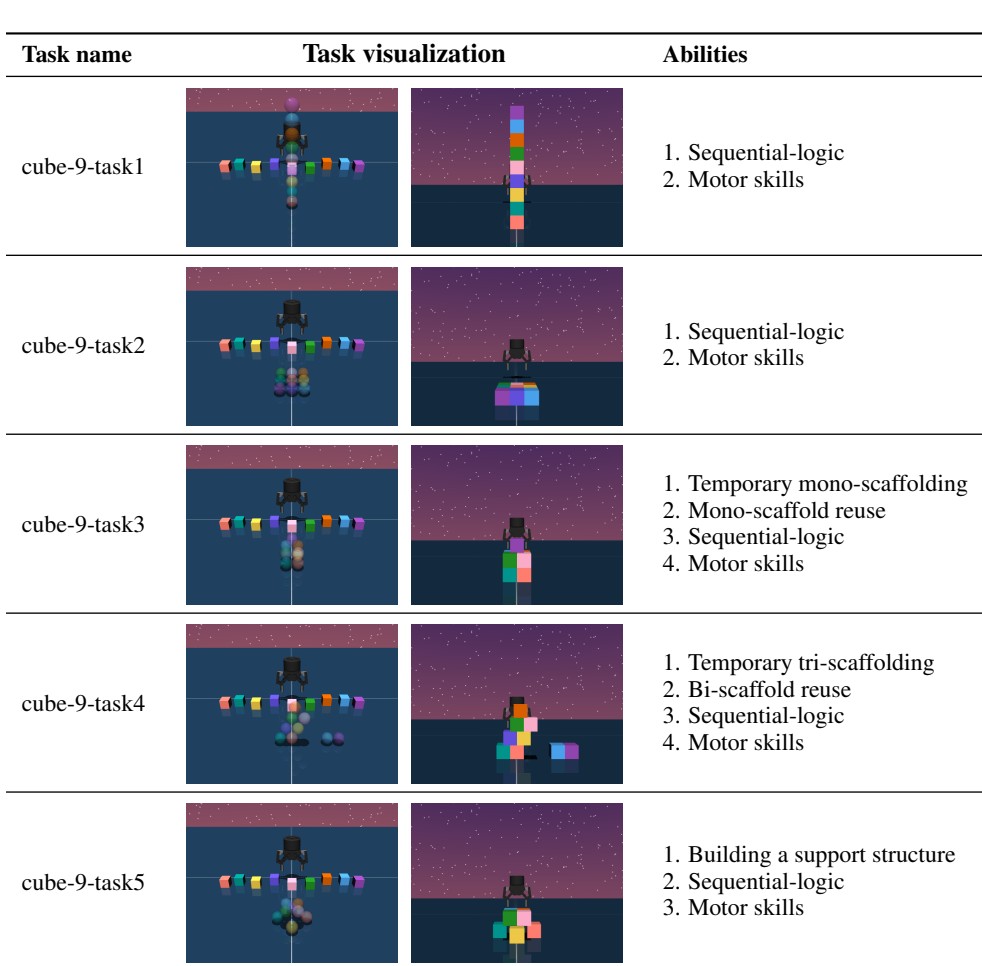 | | 1. Sequential-logic
2. Motor skills |
| cube-9-task2 | | | 1. Sequential-logic
2. Motor skills |
| cube-9-task3 | | | 1. Temporary mono-scaffolding
2. Mono-scaffold reuse
3. Sequential-logic
4. Motor skills |
| cube-9-task4 | | | 1. Temporary tri-scaffolding
2. Bi-scaffold reuse
3. Sequential-logic
4. Motor skills |
| cube-9-task5 | | | 1. Building a support structure
2. Sequential-logic
3. Motor skills |

Table 10: **Tasks with 9 cubes**.

