# OpenReview forum: "BuilderBench -- A benchmark for generalist agents"
_ICLR.cc/2026/Conference — Submitted to ICLR 2026_

### Official Review · Reviewer_9Evh · 2025-10-23

**Soundness:** 3
**Presentation:** 3
**Contribution:** 1
**Rating:** 2
**Confidence:** 2

**Summary:**

With the aim of providing a more realistic and effective RL benchmark than existing ones, enabling open-ended learning of agents in reward-free contexts (unsupervised RL), the authors propose BuildBench. This environment allows for the implementation of environments, as well as providing a suite of 50 environments to force the agent to develop reasoning skills.

**Strengths:**

S1. Apparently, it seems that AGI could come about through open-endedness, and therefore, implementing benchmarks that enable such types of learning is obviously relevant.

S2. The document is very comprehensive and provides extensive bibliography, allowing the author to clearly identify the need for BuildBench.

**Weaknesses:**

W1. The paper fails to accurately determine the limitations that exist in similar proposals in the literature, and which ones are satisfactorily resolved by BuildBench.

W2. Continuing with the previous thread, one reference published in ICML 2025 was Craftium, which was also published in RLC 2025. This platform is highly developed and allows the creation of environments for open-ended learning, procedural benchmark generation, and, in addition to multi-tasking (for CRL), it also allows for multi-agent learning.
As I see it, Craftium's scope goes beyond that of BuildBench. Not including the most recent work in the comparison undermines this study.

**Questions:**

Q1. What are the differences between Craftium and BuildBench? Does BuildBench include the capabilities offered by Craftium? What other possibilities does it offer?

---

> ### Author Response · Authors · 2025-11-20
> **Reply by Authors**
>
> We thank the reviewer for their detailed feedback.
>
> > determine the limitations that exist in similar proposals in the literature
>
> We would like to point that we discuss prior benchmarks like PhysBench, Intphys Rlbench, OpenAI Gym, Planbench and Minerva  in the related work section. We also extensively discuss various drawbacks of prior benchmarks like Kinetix, Xland and Minecraft (see last paragraph of related work section) that are fixed by BuilderBench.
>
> In Appendix C and Table 1 we have added a comparison of BuilderBench with many (including the ones mentioned by the reviewer) prior benchmarks. We will summarize the main highlight below:
>
> * Task-suite:  We have provided a task suite which
> requires skills such as reasoning about commutativity and associativity of pick and place ordering, maximizing overhangs, packing problems, intuitive physics, counterweights, buttresses, and
> temporary scaffolding. **Building such tasks is non trivial and essential for evaluating zero shot reasoning from scratch.**
>
> * Fast interaction **(3x-25x)** : Training agents to solve these tasks will presumably require large amounts of interaction. Hence fast simulators are necessary. Even if our tasks are replicated in other hardware accelarated simulators, scaling the number of objects in a scene greatly slows the simulation down. This is a well known problem. We significantly reduced this speed drop using a combination of MuJoCo's cpu threading and end to end jitting, similar to EnvPool XLA. To show this, we have added a simple speed test of BuilderBench with ManiSkill, Genesis and MuJoCo MJX when simulating 10 cubes and a robot:
>
> | Simulator  | Simulation Speed with 10 cubes (Frames per Second) |
> |---------|----------|
> | BuilderBench     |  312461  |
> | MuJoCo MJX     | 12376   |
> | Genesis | 18466 |
> | Maniskill | 98084 |
>
> These comparisons clearly highlight the limitations that exist in similar proposals in the literature shows the need for BuilderBench.
>
> > Craftium
>
> We thank the reviewer for pointing this work out. While we have cited Minecraft, we weren't aware of the Craftium project. We have added this both to the related work section and Appendix C and Table 1. We note several points that make BuilderBench particularly appealing for research on open-ended exploration, zero shot reasoning, and learning purely from trial and error
>
> * Task-suite: We have provided a task suite which requires skills such as reasoning about commutativity and associativity of pick and place ordering, maximizing overhangs, packing problems, intuitive physics, counterweights, buttresses, and temporary scaffolding. Building such tasks is non trivial and essential for evaluating zero shot reasoning from scratch.
>
> * Distinct Physics Domain: BuilderBench focuses on tasks requiring continuous physics reasoning (e.g., maximizing overhangs, utilizing counterweights, creating temporary scaffolding). These tasks are impossible to set up in the discrete, block-based world of Craftium.
>
> * Solving tasks whose solutions are unkown to us: In BuilderBench, we provide some stable final structures which we don't know how to stably build. Such tasks are not framed in Craftium. These are exactly the tasks we want machine learning models to solve. BuilderBench provides a concrete experimental instantiation of such tasks.
>
> * Most tasks in minecraft / craftium are programmable -- that is, one could write a program to solve them (killing monsters, searching for diamonds etc.). But there are many tasks in BuilderBench who's solutions do not seem to have simple programs (high Kolmogorov complexity). For instance scaffolding or generating counterweights.
>
> * Learning purely from trial and error: Training agents to solve such complex tasks will presumably require large amounts of interaction. Hence fast simulators are necessary. Craftium is not hardware accelerated. Based on Figure 7 in the Craftium paper, the BuilderBench simulator is ~400 times faster. This boost will drastically increase the speed of algorithmic iteration.
>
> **We believe the new detailed comparisons in the revised manuscript and the clear differentiation from Craftium, directly address the reviewer's concerns regarding the limitations of prior work and the novelty of our contribution. We are confident that BuilderBench provides a necessary and distinct benchmark for open-ended exploration and zero shot learning.**

---

> > ### Comment · Reviewer_9Evh · 2025-11-26
> >
> > Thank you for your  response. However, and this is something I mentioned in my review, which has been overlooked in your response, BuildBench does not support multi-agent learning. In this regard, it lags behind proposals that have already been published in machine learning literature.
> >
> > As far as I can see, BuildBench is still just a collection of scenarios (challenging), that is, a benchmark. Meanwhile, some proposals in the literature are entire frameworks that allow you to design the scenarios you believe are necessary for your research and in the learning paradigms you consider relevant. In that sense, BuildBench seems more limited to me.

---

### Official Review · Reviewer_KJoU · 2025-10-23

**Soundness:** 2
**Presentation:** 3
**Contribution:** 3
**Rating:** 6
**Confidence:** 4

**Summary:**

The paper introduces BuilderBench, a new benchmark with controllable physics-based setup for evaluating reinforcement learning (RL) agents in block stacking. Two training and evaluation protocols are proposed:
I. a multi-task self-supervised protocol, where agents learn by autonomous exploration without explicit rewards;
II. a single-task supervised protocol, where agents are trained with task-specific rewards.
The benchmark is open-sourced and includes reproducible setups and a broad set of baseline evaluations. Experiments show that while current agents perform reasonably well on simple tasks, they almost all fail on more complex structures, suggesting that current RL methods lack physical reasoning and compositional planning abilities.

**Strengths:**

1. The paper focus on testing the ability of generalization and learning of general principle, which is an important problem in RL.
2. The environment design is clean, interpretable, scalable, and well-controlled, allowing for clear attribution of success and failure.
3. The two complementary protocols make the benchmark relevant to multiple RL paradigms.
4. The benchmark is fast-speed and reproducible, potentially useful as a standard testbed for physical reasoning research.

**Weaknesses:**

1. The construct validity of the benchmark could be clearer—it is not fully demonstrated that success requires genuine physical reasoning rather than geometric features.
2. The analysis of failure modes is relatively shallow; it does not fully separate challenges of exploration, long-horizon credit assignment, and physical modeling.

**Questions:**

1. Could the authors clarify which aspects of the benchmark truly require reasoning about stability rather than brute-force exploration?
2. The dense reward is defined primarily by geometric distance to the target structure. Would incorporating physically meaningful quantities provide a more informative learning signal and better align the benchmark with its goal of evaluating physical reasoning?

---

> ### Author Response · Authors · 2025-11-20
> **Reply by Authors**
>
> We thank the reviewer for their detailed feedback. We appreciate the reviewer's positive comments about our environment and task design.
>
> > success requires genuine physical reasoning rather than geometric features
>
> This is precisely the point we want to make. "Reasoning" can be done in lots of languages and embodiments, and that it will look different from the "booksmarts" reasoning displayed by today's LLMs. **In Tables 3,4,5,6,7,8,9,10, we have added the different abilities that are needed to solve each of the task in the BuilderBench.** Solving these tasks require various skills which are generally classified as reasoning, for example -- solving math problems like maximum overhang, building and reusing scaffolds and using counterweights.
>
> > separate challenges of exploration and long-horizon credit assignment
>
> In online learning, these challenges are very related (see section 5.4 of [1]). Exploration is the process of discovering high return trajectories and credit assignment is the process of reliably imitating these trajectories. To disect these issues, each task in our task-suite has two types of rewards - dense and sparse. The dense rewards essentially alleviate the temporal credit assignment problem as better actions almost always get higher rewards. The results in Figure 6 and 7 use dense rewards. Hence, we posit that the main challenge is exploration. This hypothesis is further supported by two factors:
>
> 1) In the training runs, we can see that even though returns improve, success rates remain near zero.
>
> 2) We have provided scripts that record videos of agents after every iteration of training. Through manual inspection one can see that agents sometimes learn to complete tasks in using a locally optimal solution. For example an agent with a single cube learns to pick it up (https://drive.google.com/file/d/14eur4us-gwC0-IspdKxM8FQ84x1FXwvq/view?usp=sharing), but while stacking three cubes just pushes them near the target (https://drive.google.com/file/d/1dPQVOvCtRZduwYjHlUWqsYlrNp90lytd/view?usp=sharing). Adding exploration bonuses helps solve this problem (https://drive.google.com/file/d/1TdMdJ_cUwLcyxn5edMWsiTYoe565IeTs/view?usp=sharing).
>
> > truly require reasoning about stability rather than brute-force exploration?
>
> We believe it is impossible to tell whether a task is being solved using brute force exploration or reasoning. For example, If one is playing Chess against a computer, can one tell if the computer is doing brute-force search or "reasoning"?
>
> > physically meaningful quantities to provide a more informative learning signal
>
> Are you referring to the center of mass of the target structure? We are happy to add any quantities you suggest and run additional experiments. That being said, we believe that purely self-supervised agents (for example goal conditioned RL) that learn to build various intermediate target structures can take advantage of the dense feedback provided directly by the interaction data (actions and observations). We have implemented two such algorithms -- Contrastive RL and Hindsight Experience Replay. We believe that additional algorithmic techniques, especially to tackle the curse of horizon are needed to scale these approaches.
>
>
> We hope that these answers revisions address all the reviewer's concerns. We would be happy to address any other concerns of the reviewer.

---

### Official Review · Reviewer_Gwk7 · 2025-10-30

**Soundness:** 2
**Presentation:** 3
**Contribution:** 2
**Rating:** 4
**Confidence:** 3

**Summary:**

This work proposes BuilderBench, a benchmark designed to advance research on agent pre-training through open-ended, interactive exploration. It tasks agents with building diverse block structures in a simulator, requiring them to learn physics, math, and long-horizon planning without external supervision.

**Strengths:**

1. The task designs appropriately require physical understanding and spatial reasoning, demonstrating a thoughtful integration of these elements.
2. The design principles are coherent and well founded.

**Weaknesses:**

1. Developing a simulator using MuJoCo does not appear to be a substantial contribution, as there already exist numerous GPU-accelerated simulators with similar capabilities.
2. The problem setup is overly simplified—the flying gripper operates in 3D space but only rotates around the z-axis (i.e., performs strictly vertical). This simplification likely explains why the reinforcement learning (RL) agents achieve rapid learning.
3. Although the title claims that the benchmark targets generalist agents, the experiments primarily evaluate standard RL algorithms with different exploration strategies. In contrast, the large language model (LLM) evaluations do not involve actual interaction or feedback from the environment, thus failing to demonstrate open-ended exploration or learning through experience.

**Questions:**

Is there any feedback loop designed here other than merely reward functions? How can the generalist agent explore and receive experience  in the current environment?

---

> ### Author Response · Authors · 2025-11-19
> **Reply by authors**
>
> We thank the reviewer for their detailed feedback. We appreciate the reviewer's acknowledgement that our task designs require thoughtful integration of physical understanding and spatial reasoning. It seems that the reviewer’s main feedback are regarding (1) significance of contributions and similarity with existing simulators, (2) a simplified setup (3) and limited LLM evaluations.
>
> To address (1), In Appendix C and Table 1 we have added a comparison of BuilderBench with many (including the ones mentioned by the reviewer) prior benchmarks. **Our contributions include an extensive task suite (that is non trivial to design), and a simulator that is orders of magnitude faster and scalable than other similar hardware accelerated simulators (see below for new comparison) and a benchmarking of a total of 10 supervised and self-supervised RL algorithms.**
>
> For (2), we respectfully disagree with the reviewer’s comments that RL agents achieve rapid learning. **In fact, as evidenced by Figure 6 and Figure 7, both supervised and self-supervised algorithms achieve non-trivial performance only on the simplest tasks. Solving most BuilderBench tasks through trial and error remains an open problem.** That being said, we have also added versions of tasks with the complete robot to add more complexity (see video links below).
>
> For (3), Our focus is on agents (see footnote) that gather their own prior data (exploration from scratch) and use it to solve complex tasks (zero shot reasoning). This is exactly how prior work (Kinetix ICLR 2025 Oral) has used the term generalist agents. We have removed the term “generalist” from the body of the paper to avoid confusion(we will also remove it from the title in the final version). The LLM evaluations are meant to highlight how solving our tasks requires non-obvious steps of reasoning that are beyond what current models can achieve through scaling alone.
>
> > substantial contribution
>
> * Task-suite: We have provided a task suite which requires skills such as reasoning about commutativity and associativity of pick and place ordering, maximizing overhangs, packing problems, intuitive physics, counterweights, buttresses, and temporary scaffolding. Building such tasks is non trivial and essential for evaluating zero shot reasoning from scratch. Designing algorithms to solve these long horizon reasoning tasks through trial and error remains an open problem.
>
> * Fast interaction (3x-25x) : Training agents to solve these tasks will presumably require large amounts of interaction. Hence fast simulators are necessary. Even if our tasks are replicated in other hardware accelarated simulators, scaling the number of objects in a scene greatly slows the simulation down. This is a well known problem. We significantly reduced this speed drop using a combination of MuJoCo's cpu threading and end to end jitting, similar to EnvPool XLA. To show this, we have added a simple speed test of BuilderBench with ManiSkill, Genesis, MuJoCo MJX when simulating 10 cubes and a robot:
>
>     | Simulator  | Simulation Speed with 10 cubes (Frames per Second) |
>     |---------|----------|
>     | BuilderBench     |  312461  |
>     | MuJoCo MJX     | 12376   |
>     | Genesis | 18466 |
>     | Maniskill | 98084 |
>
> * Benchmarking of 10 algorithms : To increase the scope of BuilderBench, we have added benchmarked multiple new algorithms for both protocols in Figure 6 and Figure 7. The following is the list of all algorithms:
>
>     * (SSL) Upside Down RL.
>     * (SSL) Random Network Distillation.
>     * (SL) Graph Attention networks.
>     * (SL) Bigger Regularized Optimistic.
>     * (SL) Contrastive RL.
>     * (SL) Random Network Distillation.
>     * (SL) Proximal Policy Optimization.
>     * (SL) Soft Actor Critic.
>
> **We hope that these new results and clarity of contributions address the concern of the reviewer.**
>
> > RL agents achieve rapid learning
>
> As pointed above, Figure 6 and Figure 7, show that both supervised and self-supervised RL algorithms achieve non-trivial performance only on the simplest. We have added a setup with a complete robot.  See the link for a video of a trained policy https://drive.google.com/file/d/1O2mfts3la-7PRbdg61K3GnhVxZPINO5U/view.
>
> > generalist agents and LLMs
>
> We have removed occurrences of generalist agents from the body of our paper and will remove it from the titles in the final version. We would like to note that we were following prior work (Kinetix ICLR 2025 Oral) in the usage of generalist.
>
> > feedback loop other than merely reward functions? How can the generalist agent explore and receive experience?
>
> That's precisely the point of our benchmark. The only feedback loop is supposed to be trajectories of experience (without any rewards). The aim is to facilitate research on algorithms that autonomously learn about their world without any explicit feedback. While we implement algorithms (see Figure 6) which make progress on the simplest of tasks, scaling to more difficult tasks remains an open problem.

---

### Official Review · Reviewer_X9na · 2025-11-02

**Soundness:** 3
**Presentation:** 3
**Contribution:** 2
**Rating:** 4
**Confidence:** 4

**Summary:**

This paper presents a new benchmark called BuilderBench, specifically for evaluating generalization in reinforcement learning agents. This benchmark consists of around 50 open-ended block-building tasks like T-bock, 4-cube-packing, leaning tower, etc, that test an RL agent’s capacity to physically reason with cubes. The authors also discuss some design decisions behind the benchmark like the tasks requiring distinct skills, having a curriculum of easy to difficult tasks. A few algorithms, including ChatGPT 5 and Gemini 2.5 Pro are evaluated on BuilderBench in the single-task and multi-task regime. None of the models and algorithms evaluated show signs of success beyond the simplest setup. The paper concludes by stating that better algorithm design like hypothesis driven exploration and is required to solve the introduced benchmark.

**Strengths:**

1. The writing is excellent and the language is simple to follow.
2. The paper has a strong and relevant motivation to improve RL agents towards grounded reasoning and generalization.
3. The created tasks are intuitive and lie on a spectrum of easy to difficult w.r.t. current RL algorithms tested in the paper. This might help develop curriculum based learning methods.
4. The self-supervised protocol presented has the potential to push the field towards unsupervised RL research. Especially in developing hypothesis driven exploration and grounded reasoning methods for embodied agents.
5. I agree with the final takeaway message that the authors state - existing algorithms might lack the grounded reasoning and compositional abilities to solve difficult manipulation problems. This is definitely an area in RL that requires further attention and research work.

**Weaknesses:**

Also adding questions in this section:
1. While I highly appreciate the clear goal, simple presentation, and a well implemented RL environment, I would expect a benchmark paper to present more rigorous experimentation and conclusion - the current paper falls short of this expectation. From the perspective of science, implementing PPO & SAC, and stating that other algorithms are “out of scope for the paper” (Limitations & Conclusion, page 9) is simply not sufficient to conclude that existing research on generalization is weak (although I intuitively agree with the authors’ conclusions). I say this with utmost appreciation for the work by authors: Without rigorous quantitative benchmarking and ablatory analysis on scaling behavior (sample efficiency/compute cost analysis), this research contribution is just a little better than an open-source software release. The authors of a benchmark paper have the opportunity to evaluate multiple existing algorithms in a completely unbiased manner which a future work that push a particular algorithm and agenda might lack. Given the claim of simplicity and speed of this environment, I think it is essential to run experiments on other existing methods and present the results and analyses. I would like to see standard supervised RL benchmarks on other methods like TDMPC2 [1] (a model-based approach) and other approaches specifically developed for Compositional tasks and object centric behavior [2][3][4][5]. Several competent VLAs & LLM augmented models also have attempted to achieve grounded reasoning capabilities [10][11]. Unsupervised exploration is also covered by several notable prior works [6][7]. **Atleast one other algorithm that covers the breadth of research directions for each supervised and self-supervised protocol** would improve the contribution greatly.
2. Can the authors concretely specify what sets apart their benchmark? Several existing works attempt to capture compositional and logical reasoning abilities in RL and robotics with simple-to-hard curriculum. Examples: FurnitureBench [9], BabyAI [12]. If there are significant novel changes, please specify why these new changes are essential and what prompted the need for an entirely new benchmark? For example, Maniskill [8] is a well established benchmark which already contains tasks like PickYCB, YCB Clutter, Stack cube, etc. Maniskill also provides demonstrations, RGBD & segmentation mask observations. What is the utility of a new benchmark when one could add more cubes and create these tasks on the current benchmarks. Please mention clearly.
3. Are demonstrations available for the tasks in this benchmark? Prior benchmarks like Maniskill [8] and FurnitureBench [9] also provide demonstrations and evaluations on generalizable Imitation learning methods which are additionally considered important approaches in the RL and robotics community.
4. Similarly, can fast rendering of RGBD observations be made available? With the popularity of Vision Language Models and DINO style vision encoders, visual observations are an essential component for modern benchmarks.

**Questions:**

Please see above.

Overall I would consider this paper a weak reject since it lacks in experimentation and standard RL benchmark features such as RGBD observations and expert demonstrations. Authors, please answer my questions above and consider adding a few more benchmarks. I will revise my score if my assumptions are incorrect and/or the additional responses/experiments clarify my doubts and help improve the scientific contribution of this work.

**References**

*[1] Hansen, N., Su, H., & Wang, X. (2024). TD-MPC2: Scalable, robust world models for continuous control. In Proceedings of the International Conference on Learning Representations (ICLR 2024)*

*[2] Ghasemipour, et.al. (2022). Blocks Assemble! Learning to assemble with large-scale structured reinforcement learning. In Proceedings of the 39th International Conference on Machine Learning (ICML 2022) (pp. 7435–7469). PMLR*

*[3] Mishra, U. A. et.al. (2023). Generative skill chaining: Long-horizon skill planning with diffusion models. In Proceedings of the 7th Conference on Robot Learning (CoRL 2023)*

*[4] Shridhar, M., Manuelli, L., & Fox, D. (2021). CLIPort: What and where pathways for robotic manipulation. CoRL 2021*

*[5] Zeng, A., et al. (2021). Transporter networks: Rearranging the visual world for robotic manipulation. CoRL 2020*

*[6] Laskin, M., et al. (2021). URLB: Unsupervised Reinforcement Learning Benchmark. NeurIPS Datasets & Benchmarks*

*[7] Ecoffet, A., et al. (2021). First return, then explore (Go-Explore). Nature*

*[8] Gu, S., et.al. (2023). ManiSkill2: A unified benchmark for generalizable manipulation skills. arXiv preprint arXiv:2302.04659*

*[9] Nair, S., et.al. 2022). FurnitureBench: Reproducible real-world benchmark for long-horizon complex manipulation. Proceedings of Robotics: Science and Systems (RSS).*

*[10] Kalithasan, N., et al. (2024). Sketch-Plan-Generalize: Learning and planning with neuro-symbolic programmatic representations for inductive spatial concepts. arXiv preprint arXiv:2404.07774*

*[11] Ahn, M., et al. (2022). Do As I Can, Not As I Say (SayCan): Grounding language in robotic affordances. arXiv:2204.01691*

*[12] Chevalier-Boisvert, M., et al. (2019). BabyAI: A platform to study the sample efficiency of grounded language learning. International Conference on Learning Representations (ICLR)*

---

> ### Author Response · Authors · 2025-11-19
> **Reply by authors**
>
> We thank the reviewer for their detailed feedback. It seems like the reviewer's main concerns are regarding (1) breadth of algorithms that are benchmarked and (2) clarity on features of BuilderBench that set it apart.
>
> To address (1), **In Figure 6 and Figure 7 we have added 4 new algorithms**, each covering a different direction. Self supervised protocol: Upside Down RL, Random Network Distillation. Supervised protocol: Contrastive RL, Bigger Regularized Optimistic, Graph Attention networks, Random Network Distillation.
>
> To address (2), In Appendix C and Table 1 we have added a comparison of BuilderBench with many (including the ones mentioned by the reviewer) prior benchmarks. **This comparison clearly shows the need for BuilderBench.**
>
> **Do these answers, revisions and experiments address all the reviewer's concerns?**
>
> > breadth of research directions
>
> To increase the scope of BuilderBench, we have added multiple new algorithms for both protocols in Figure 6 and Figure 7:
>
> * (SSL) Upside Down RL - self supervised algorithms that only learns a mapping from goals to actions using hindsight relabelling.
> * (SSL) Random Network Distillation -- self supervised algorithms that explores just using an exploratory intrinsic reward.
> * (SL) Graph Attention networks -- RL algorithm that uses a graph attention network similar to [1].
> * (SL) Bigger Regularized Optimistic -- RL algorithm that uses the BRO network proposed by [2].
> * (SL) Contrastive RL -- GCRL algorithm that solves the RL problem estimates values using contrastive learning.
> * (SL) Random Network Distillation -- RL algorithm that uses a mix of extrinsic and intrinsic rewards.
>
> BRO is a SOTA algorithm which has been shown to outperform TD-MPC 2. Graph Attention networks were especially proposed for tasks where the agents are compositional in nature. While these algorithms focus on sample efficiency, we find that they do not necessarily outperform PPO when scaled to thousands of parallel environments and more complex tasks. We argue that these new algorithms and results increase the feasibility of research that can be done using BuilderBench.
>
> > What sets BuilderBench apart?
>
> In Appendix C and Table 1 we have added a comparison of BuilderBench with many (including the ones mentioned by the reviewer) prior benchmarks. We will summarize the main highlight below:
>
> * Task-suite:  We have provided a task suite which
> requires skills such as reasoning about commutativity and associativity of pick and place ordering, maximizing overhangs, packing problems, intuitive physics, counterweights, buttresses, and
> temporary scaffolding. **Building such tasks is non trivial and essential for evaluating zero shot reasoning from scratch.**
> * Fast interaction **(3x-25x)** : Training agents to solve these tasks will presumably require large amounts of interaction. Hence fast simulators are necessary. Even if our tasks are replicated in other hardware accelarated simulators, scaling the number of objects in a scene greatly slows the simulation down. This is a well known problem. We significantly reduced this speed drop using a combination of MuJoCo's cpu threading and end to end jitting, similar to EnvPool XLA. To show this, we have added a simple speed test of BuilderBench with ManiSkill, Genesis and MuJoCo MJX when simulating 10 cubes and a robot:
>
> | Simulator  | Simulation Speed with 10 cubes (Frames per Second) |
> |---------|----------|
> | BuilderBench     |  312461  |
> | MuJoCo MJX     | 12376   |
> | Genesis | 18466 |
> | Maniskill | 98084 |
>
> > demonstrations for the tasks
>
> In our code, we have provided a planner that can solve a minority of tasks which only require pick and place primitives. Researchers can generate demos using this policy. But most tasks are so hard that we don't know how to generate policies that solve them --- that's precisely the problem we want to solve.
>
> > VLAs & LLM augmented models & human designed curriculum and prior data
>
> While we agree that this is an exciting direction, BuilderBench is meant to focus on agents that learn purely through trial and error (see introduction and related works). *We would like to point that Kinetix [3], a similar but 2-dimensional benchmark, was accepted as an oral at ICLR 25' without any of these comparisons.*
>
> > fast rendering of RGBD for Vision Language Models and DINO like encoders
>
> We would like to note again that BuilderBench is meant to focus on agents that learn purely through trial and error, similar to Kinetix [3]. That being said, a single BuilderBench environment can be rendered and used for training. Future work could certainly incorporate a high-speed rendering extension (e.g., using a renderer like Madrona).
>
> [1] Blocks Assemble! Learning to assemble with large-scale structured reinforcement learning.
>
> [2] Bigger, Regularized, Optimistic: scaling for compute and sample-efficient continuous control.
>
> [3] Kinetix: Investigating the Training of General Agents through Open-Ended Physics-Based Control Tasks.

---

### Meta-Review · Area_Chair_fRYV · 2025-12-29

**Summary:**

The paper presents BuilderBench, a benchmark for agents to stack cubes in a physically plausible way without toppling.  The benchmark consists of 42 tasks, designed with a range of difficulties, and is built on top of the MuJuCo physics simulator.  Experiments are conducted on the proposed benchmark to evaluate the performance of different RL agents and LLMs for planning.

While reviewers appreciated the attempt to create an interesting benchmark for block stacking, reviewers expressed 1) concerns about the limited coverage of RL methods and LLMs in the benchmarking effort and 2) the unclear value of the proposed tasks in relation to other existing benchmarks.  Due to the above, some reviewer also expressed an opinion that the contribution of the work is not sufficient.

While the authors attempted to address these concerns, the AC does not believe the concerns to be sufficiently addressed.  The value of the proposed benchmark is unclear, and as the scope of the experiments is limited, the conclusions and findings from it does not seem to be of sufficient interest to the community.  The AC does not recommend the work be accepted at ICLR.

**Reviewer Concerns:**

Reviewer had the following concerns:
1. Experiments have limited coverage of different RL methods [X9na] and LLM evaluations do not interact with environment [Gwk7]
2. Questions about how the proposed benchmark relate to existing benchmarks such as FurnitureBench, ManiSkill, and others [X9na, 9Evh]
3. Questions about the task setup and information included in the benchmark [Gwk7, KJoU]
   a. Oversimplified setup with overhead gripper [Gwk7]
   b. Unclear that physical reasoning is required to solve the tasks [KJoU]
   c. Questions about whether demonstrations are available for the tasks [X9na]
4. Insufficient contribution [Gwk7]
5. Shallow analysis of failure modes [KJoU]
6. Question about the rendering speed of RGBD observations [X9na].
7. Lack of multi-agent support [9Evh]

The AC believes the key concerns are the limited benchmarking (1), questions about how the benchmark relates to prior work (2), and questions about the task setup (3).  During the author response period, the authors added experimental coverage for 4 more RL methods (bringing up the total to 10 different RL methods), and indicated that one of the key differences to prior work was the fast simulation with 10 cubes.  There is little information as to what improvements the proposed work did over MuCoJo to achieve the fast simulation.

The paper should situate itself better with respect to prior work that focused on 1) using stacking for physical stability and 2) other simulation based assembly benchmarks, as well as 3) provide improved arguments about why stacking is a important task, and in what ways does it require for physical reasoning (is it actually reasoning that is required, or just physical awareness?), 4) improvements over Mujoco that accounts for the fast simulation performance.  If physical reasoning is a main focus, then the AC also recommends situating the work with respect to non-stacking / assembly based physics reasoning benchmarks.

Minor comments:
- Typo: L977 => "BuilerBench" => "BuilderBench"
- If there is a section 7.1, there should be section 7.2
- The term "generalist" is confusing and not completely removed from the paper (although the authors claim to have 'removed the term “generalist” from the body of the paper'). Other than the title, the term is still in Figure 1 and in the section three heading.

**Reviewer Scores:**

Reviewers were mostly negative on this work, with 1 reject (9Evh), two marginal accepts (X9na, Gwk7) and one marginal accept (KJoU).

Although the authors responded to reviewers by Nov 19th, reviewers did not engage in discussion (with the exception of R-9Evh).  There was no indication from any reviewers that the author response has convinced the reviewers to change their scores.

---

### Decision · Program_Chairs · 2026-01-26

Reject